# Evolutionary trade-off and mutational bias could favor transcriptional over translational divergence within paralog pairs

**Simon Aubé**[1,2,3,4]*, **Lou Nielly-Thibault**[2,3,4,5¤], **Christian R. Landry**[1,2,3,4,5]*

**1** Département de biochimie, de microbiologie et de bio-informatique, Faculté des sciences et de génie, Université Laval, Québec, Québec, Canada, **2** Institut de Biologie Intégrative et des Systèmes, Université Laval, Québec, Québec, Canada, **3** PROTEO, Le regroupement québécois de recherche sur la fonction, l'ingénierie et les applications des protéines, Université Laval, Québec, Québec, Canada, **4** Centre de Recherche en Données Massives, Université Laval, Québec, Québec, Canada, **5** Département de biologie, Faculté des sciences et de génie, Université Laval, Québec, Québec, Canada

¤ Current address: Département de médecine moléculaire, Faculté de médecine, Université Laval, Québec, Québec, Canada
* simon.aube.2@ulaval.ca (SA); christian.landry@bio.ulaval.ca (CRL)

**Data Availability Statement:** All data preparation, simulations and analysis were done in Python (v 3.10 for simulations and v 3.8.16 for analyses done in notebooks), with all statistical tests being

## Abstract

How changes in the different steps of protein synthesis—transcription, translation and degradation—contribute to differences of protein abundance among genes is not fully understood. There is however accumulating evidence that transcriptional divergence might have a prominent role. Here, we show that yeast paralogous genes are more divergent in transcription than in translation. We explore two causal mechanisms for this predominance of transcriptional divergence: an evolutionary trade-off between the precision and economy of gene expression and a larger mutational target size for transcription. Performing simulations within a minimal model of post-duplication evolution, we find that both mechanisms are consistent with the observed divergence patterns. We also investigate how additional properties of the effects of mutations on gene expression, such as their asymmetry and correlation across levels of regulation, can shape the evolution of paralogs. Our results highlight the importance of fully characterizing the distributions of mutational effects on transcription and translation. They also show how general trade-offs in cellular processes and mutation bias can have far-reaching evolutionary impacts.

## Author summary

Changes in the cellular abundance of proteins are of great importance in evolution, as they are associated with phenotypic variation and adaptation. They can result from mutations acting on multiple biochemical processes, of which the most important are the transcription of mRNAs and their translation into polypeptides. While the evolution of transcription levels has been extensively studied, the interplay between transcriptional and translational changes remains to be fully elucidated. Yet, there is accumulating evidence that transcription may evolve at a faster rate. We show that this is the case within paralog pairs of the yeast *Saccharomyces cerevisiae*, where divergence is significantly larger

performed using their SciPy (v 1.8.0 for simulations and v 1.7.3 for analyses done in notebooks) implementation. The corresponding code, as well as all command lines used to perform the simulations, have been deposited on GitHub: https://github.com/Landrylab/Aube_et_al_2022. The minimal dataset necessary to replicate all figures is also included in this repository. All primary simulation data can be generated using the provided code, which specifies all parameter values and random seeds used.

**Funding:** Funding for this work was from a NSERC discovery grant (RGPIN-2020-04844) to CRL. SA was supported by graduate scholarships from NSERC and FRQNT, as well as a PBLDD scholarship from Université Laval. LNT was supported by an Alexander Graham Bell doctoral scholarship from NSERC. CRL holds the Canada Research Chair in Cellular Systems and Synthetic Biology. The funders had no role in study design, data collection and analysis, decision to publish, or preparation of the manuscript.

**Competing interests:** The authors have declared that no competing interests exist.

at the transcriptional level than at the translational one. Using simulations, we compare two potential mechanisms by which such patterns could arise: an evolutionary trade-off in the process of protein synthesis and a higher probability—or larger effects—for mutations affecting transcription. We find that both explanations are consistent with the observed divergence of duplicated genes. Whether these apply generally to all genes or exclusively to paralogs, this work still provides important insights on the evolution of gene expression levels. Gene duplication events indeed occur frequently and are thus likely to have profound and lasting impacts on the evolution of protein abundance levels.

## Introduction

Gene expression level—the steady-state cellular abundance of the corresponding protein—is a fundamental property of genes, as shown by extensive reports of fitness-expression dependencies across organisms and biological functions [1–3]. As such, understanding its evolution is of great biological importance. Gene expression is a multi-step process, involving the transcription of mRNAs and their translation into proteins, as well as the active degradation of both types of molecules and their dilution during cell division [4]. Accordingly, variation in protein abundance among genes within or between species can arise from changes at multiple levels. While many studies have described the crucial role of mutations within genes, their regulatory sequences and their regulators in generating expression variation, most reports focused on one of these aspects, for instance on the transcriptional component [5]. How each of these levels of regulation change during evolution, independently or jointly, thus remains to be fully elucidated.

One intriguing possibility which has emerged following recent investigations is that transcription may evolve at a higher rate than translation, such that variation in the abundance of transcripts would accumulate faster than changes in their translation efficiencies. It has for instance been reported that expression divergence between humans and other primates occurred mostly at the level of mRNA abundance, with little translational contribution [6]. Similarly, virtually none of the expression divergence observed between lines of the bacterium *Escherichia coli* evolved for 50 000 generations involved changes in the translation efficiency of transcripts [7]. A similar observation has been made in yeasts, where a greater number of variants affecting the abundance of mRNAs rather than their translation have been identified between *Saccharomyces cerevisiae* strains [8]. Further interspecies comparisons however revealed mostly equal contributions of transcriptional and translational changes [9–11]. Extensive reports that changes in transcription are only partially masked by variations in translation might also support a higher evolutionary rate for transcription. Such observations have been made both in mammals [12, 13] and in yeasts [9–11], although the latter are more ambiguous and have been challenged [8, 14].

Overall, it appears likely that transcriptional changes play a larger role than translational ones in the evolution of gene expression levels. Potential mechanisms underlying this discrepancy however remain to be elucidated. One powerful context in which such an investigation can be performed is that of gene duplication, an evolutionary process which creates a pair of paralogs from an ancestral gene. Since the two resulting paralogs are usually identical, their expression levels are likely similar immediately after the duplication event. Paralogs would thus gradually diverge from a common starting point over millions of years of evolution by the accumulation of expression changes. As such, differences in transcription and in translation which can currently be measured between two paralogs allow to approximate their relative

evolution in both dimensions. Moreover, the numerous paralog pairs present in a given organism constitute as many evolutionary replicates within the same cell environment. Because a given variation in protein abundance can be obtained from an infinity of transcriptional and translational changes, any consistent pattern across paralog pairs may be telling about the underlying mutational process as well as the selective pressures potentially involved. Most importantly, a predominant role of changes at the level of mRNA abundance in the expression divergence of paralogs has previously been reported in two model plant species [15].

In addition to providing a model for the study of the evolution of gene expression levels, the divergence of paralogs is in itself of high biological relevance. It is estimated that between 30% and 65% of all genes are part of duplicate families in most eukaryotes [16, 17], while new single-gene duplications may be more frequent than single-nucleotide mutations [18, 19]. Besides their high frequency, gene duplication events also have far-reaching consequences. This phenomenon is often associated with the divergence of the resulting paralogs into two functionally distinct genes through processes known as neofunctionalization and subfunctionalization, respectively involving the acquisition of new function(s) [20] and the partitioning of ancestral ones [21]. Protein abundance changes frequently accompany and may even shape this divergence. Post-duplication expression reduction has for instance been reported within paralog pairs [22]. Moreover, a compensatory drift of expression levels may also occur, allowing both gene copies to diverge while maintaining a constant cumulative protein abundance [23, 24]. As such, elucidating how the transcription and translation of paralogous genes jointly evolve is important to better understand both the general evolution of gene expression levels and the evolutionary impact of gene duplications.

To this end, we leveraged a published set of transcriptional and translational measurements for 4440 genes of the yeast *S. cerevisiae* [25], which is a well-recognized model for the study of gene duplication. These data present a major advantage: they are transcription and translation rates expressed in molecular terms, respectively transcripts synthesized per time unit and proteins produced per transcript per time unit. They thereby allow for the investigation of potential mechanisms at the molecular level. We first validated whether the paralog pairs included in this dataset show a significantly larger divergence in transcription than in translation. Having confirmed this observation, we next investigated potential underlying mechanisms. We considered two hypotheses: an evolutionary trade-off in the optimization of gene expression levels [25] and the possibility that transcriptional mutations are more frequent and/or have larger effects. Using *in silico* evolution, we show that both explanations are consistent with the observed divergence patterns. We conclude by stressing the need for measuring the mutational parameters of genes in transcription and in translation in order to be able to further support one model or the other, as well as to fully understand the multi-level evolution of gene expression.

## Results

### Yeast paralogs mostly diverged in transcription

We first examined whether transcription changes played a larger role in the evolution of yeast paralogs by comparing the extent of transcriptional and translational divergence within duplicate pairs using transcription rates $\beta_m$ and translation rates $\beta_p$—respectively in mRNAs per hour and in proteins per mRNA per hour [25]. Because protein abundance is proportional to the product of these two rates, they contribute equally to overall expression and their relative changes are directly comparable.

Among the 4440 genes for which $\beta_m$ and $\beta_p$ have been estimated, we identified 409 high-confidence paralog pairs, each derived from a whole-genome duplication (WGD; n = 245) or a single small-scale duplication event (SSD; n = 164). We quantified the contributions of

transcriptional and translational changes by computing the magnitude of relative divergence in transcription and translation as follows, where $\theta$ represents the transcription or translation rates of paralogs 1 and 2 within a pair:

$$log_2\text{-fold change} = log_2\left(\frac{max\left(\theta_1, \theta_2\right)}{min\left(\theta_1, \theta_2\right)}\right) \quad (1)$$

The distributions of these two measures across paralog pairs are consistent with a higher evolutionary rate for the regulation of transcription than for that of translation, as relative divergence in $\beta_m$ is significantly higher (Fig 1A). The median is 1.5 times larger in transcription than in translation, with only slight differences depending on duplication type (1.45× for WGD and 1.48× for SSD). In addition, about 70% (74.3% for WGD and 62.2% for SSD) of paralog pairs are more divergent transcriptionally than translationally. A potential caveat is that the transcription rates used have been obtained under the assumption that decay rates do not vary among transcripts [25]. When measurements of mRNA decay [26] are employed to recalculate $\beta_m$, the observation that relative divergence is larger in transcription than in

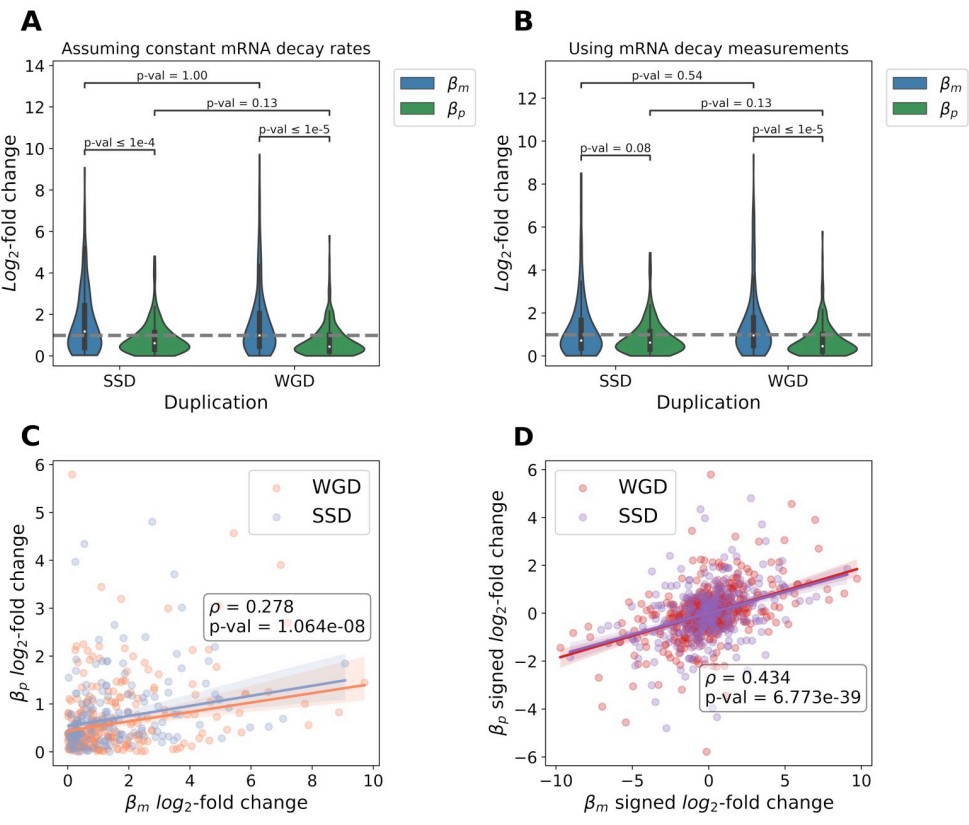

**Fig 1. Gene expression divergence of duplicated genes is greater in transcription than in translation.** (A) Distributions of relative divergence for *S. cerevisiae* paralog pairs according to the associated mechanism of duplication (245 WGD pairs; 164 SSD pairs), using transcription and translation rates inferred by [25]. The dashed line indicates the median transcriptional divergence of WGD pairs. P-values from Mann-Whitney-Wilcoxon two-sided tests are shown. (B) Same distributions of relative divergence, but using transcription rates recalculated to account for gene-to-gene variations in mRNA decay [26]. The dashed line is set to the same value as in A. (C) Correlation (Spearman's $\rho$) between the magnitudes of relative divergence in transcription and translation rates across all paralog pairs. (D) Correlation (Spearman's $\rho$) between the signed relative divergences in transcription and in translation for all duplicate pairs. Signed $log_2$-fold changes were calculated in the two possible orientations for each gene pair and the correlation was computed on the resulting duplicated dataset (n = 818).

translation holds, although statistical significance is lost for SSD-derived paralog pairs (Fig 1B). Further analyses using three other independent sets of mRNA decay rates [27–29]—obtained through distinct experimental approaches—confirm our initial observation. In all cases, the relative expression divergence is significantly higher in transcription than in translation ($p < 0.05$; Mann–Whitney–Wilcoxon two-sided test) for WGD- and SSD-derived paralog pairs, although with different magnitudes (S1B Fig). We also made sure that $\beta_p$ $log_2$-fold changes are representative of the true translational variation between paralogs. To this end, we compared the protein abundance $log_2$-fold changes calculated from $\beta_m$ and $\beta_p$ divergence to experimental measurements of protein abundance, which revealed strong correlations ($r = 0.63$- $0.80$; S1E Fig).

An additional potential confounder is experimental variation. If transcriptional measurements were noisier than translational ones, the magnitude of relative divergence in transcription could be artificially inflated. This is however unlikely to be the case, owing to how transcription and translation rates were inferred from mRNAseq and ribosome profiling data [30]. Whereas $\beta_m$ was obtained through a normalization of the mRNA abundances measured by sequencing, $\beta_p$ for a given gene was defined as the normalized ratio of its number of ribosomal footprints over its mRNA abundance [25]. Hence, translation rates could be inherently noisier, since they compound any noise in mRNAseq measurements affecting $\beta_m$ as well as additional variance introduced by the ribosome profiling. We could thus expect our assessment of expression divergence to be conservative regarding any predominance of transcriptional changes. The analysis of simulated data supports this intuition, as the addition of noise leads to an underestimation of the relative contribution of transcription in most cases (S1 Fig). The robustness of our initial observation is further confirmed by additional simulations combining the effect of experimental noise with that of unaccounted-for variations in transcript decay rates. While slightly overestimating the relative contribution of transcription changes appears plausible, in no instance does a predominantly translational divergence falsely appear mostly transcriptional (S3 Fig).

To more thoroughly characterize the joint evolution of transcriptional and translational regulation within yeast paralog pairs, we also looked for correlations between the relative magnitudes of divergence in transcription and in translation. This revealed a weak but significant positive association (Fig 1C and S1C Fig). We additionally repeated this analysis without first defining the log-fold changes as strictly positive, thus preserving information on the direction of the changes at both levels. In this case, a stronger positive correlation is observed (Fig 1D and S1D Fig). Because mRNA abundances are used in the calculation of both $\beta_m$ and $\beta_p$ [25], spurious correlations between the magnitudes of relative divergence may occur (S4A Fig). Taking into account the very strong association between the mRNA abundance and ribosomal footprints of individual genes within the dataset ($r = 0.981$, $p < 1 \times 10^{-6}$; S4B Fig), only very weak to nonexistent correlations are however expected between the magnitudes of relative divergence, both for absolute and signed values (S4C Fig). Accordingly, the relationships observed here suggest that duplicate pairs which diverged more transcriptionally also tend to have accumulated more changes at the level of translation—and usually in the same direction. This finding may reflect the action of gene-specific selective constraints on protein abundance or the existence of a correlation between transcriptional and translational mutations (see below).

Interestingly, a recent study examining the divergence of mRNA abundance and translation efficiency (analogous to transcription and translation rates, respectively) within paralog pairs in the model plants *Arabidopsis thaliana* and *Zea mays* reported patterns strikingly similar to our observations (Fig 1), with one notable exception [15]. In this study, a predominance of compensation between changes at the transcriptional and translational levels was observed,

disagreeing with the positive correlation that we report between signed divergences in transcription and translation (Fig 1D). This discrepancy is consistent with predicted differences in the efficiency of selection, and may support an involvement of selective constraints. The effective population size of *A. thaliana* is indeed one to two orders of magnitude smaller than that of *Saccharomyces* yeasts [31–33]. These organisms however differ in many other ways, as plants are multicellular—meaning that paralogs could also diverge along other dimensions that are not captured in yeasts—and the regulation of their gene expression may feature additional layers of complexity.

## A minimal model of post-duplication expression evolution

We next investigated how a higher evolutionary rate for the regulation of transcription than for that of translation—as seen among yeast paralogs—could potentially emerge. To this end, we considered two non-mutually exclusive hypotheses (Fig 2A). First, transcriptional changes might accumulate faster because they have more beneficial—or less deleterious—fitness effects than translational ones. Such a discrepancy could arise through a recently described evolutionary trade-off [25], according to which corresponding changes in transcription and in translation are equivalent in terms of amount of protein produced, but not in cost and precision (Fig 2A, left). Accordingly, the precision and economy of gene expression cannot be maximized simultaneously, although it would provide fitness benefits, because they inversely depend on the relative contributions of transcription and translation. While a larger transcriptional contribution reduces the magnitude of stochastic fluctuations in protein abundance [35, 36]—or expression noise—and thus increases precision, it also incurs additional metabolic costs through the synthesis of more mRNA molecules [25, 37, 38]. Within this framework, the sole relevant cost of gene expression is this latter transcription-dependent one, as the cost of protein production itself only depends on how many molecules are synthesized, regardless of whether translation is done from many or few mRNAs. A greater relative translational contribution on the other hand increases economy (since fewer transcripts are synthesized) but amplifies noise, which thereby decreases precision. By having distinct effects on the economy and precision of gene expression, changes to $\beta_m$ and $\beta_p$ could therefore be differently favored by natural selection.

Second, the faster accumulation of transcriptional variation could be explained without any direct involvement of selection, if the rate of transcription was more likely to be altered by mutations. Our other hypothesis is thus that transcription has a larger mutational target size than translation (Fig 2A, right), meaning that mutations acting on this trait would be more frequent or have larger effects, or both. Under neutral evolution or even under stabilizing selection to maintain protein abundance, more changes would thereby accumulate at the transcriptional level.

In order to test these two hypotheses (precision-economy trade-off vs differences in mutational target sizes), we defined a minimal model of post-duplication evolution. Within this framework, natural selection solely acts to maintain the cumulative expression of a pair of duplicated genes at a certain level (S1 File), as the two proteins are functionally equivalent. Such selection on total protein abundance is likely an important feature of the early evolution of most paralog pairs, as suggested by several observations. The retention of duplicates after WGD events is for instance higher for genes whose products are part of protein complexes [39–41], for which the lowered expression resulting from the loss of a gene copy would be more deleterious. In addition, reduction in the cumulative expression of paralog pairs, so as to more closely replicate the expression of their singleton ancestor, appears widespread [22]. More importantly, *quantitative subfunctionalization*—under which selection acts on the

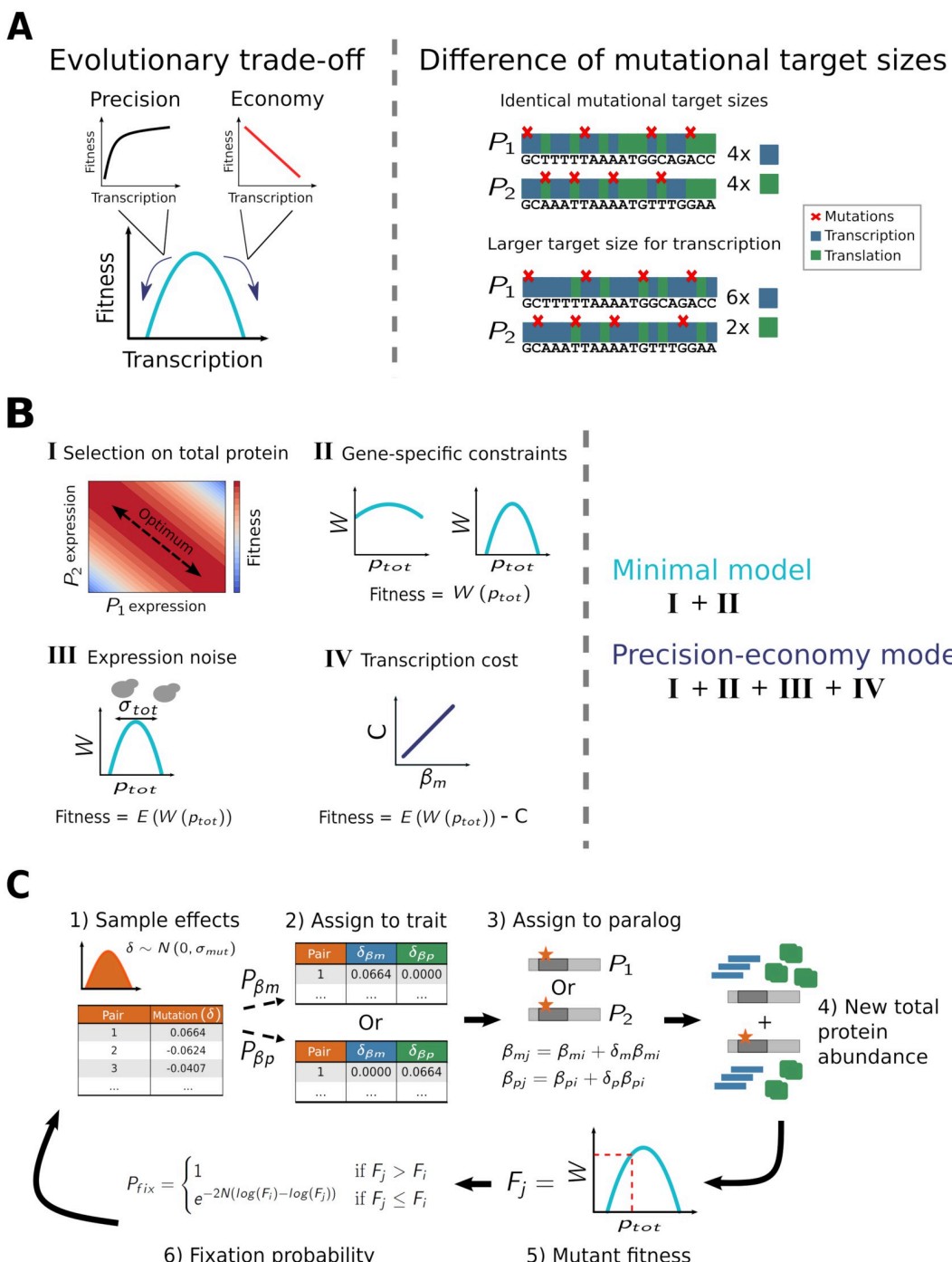

**Fig 2. Overview of the modeling and simulation approach.** (A) Two hypotheses are considered: an evolutionary trade-off between the precision and economy of gene expression (left) and a larger mutational target size for transcription than for translation (right). (B) A minimal model of post-duplication evolution is defined under the assumption that selection acts to maintain the cumulative protein abundance of two paralogs. Parabolic functions of fitness according to cumulative protein abundance with varying curvatures are used for different gene pairs. The model is complexified by the addition of precision-economy constraints [25] to test the hypothesized evolutionary trade-off. (C) The expression divergence of paralogs is simulated by sequential fixation, through multiple rounds of mutation-selection. A relative mutational effect $\delta$ is first sampled from a normal distribution of mean 0 and standard deviation $\sigma_{mut}$ for each paralog pair. It is then assigned to transcription or translation rate, according to relative probabilities $P_{\beta m}$ and $P_{\beta p}$. This mutation is randomly performed on one of the two paralogs $P_1$ and $P_2$, which are equally likely to be mutated. The transcriptional ($\delta_{\beta m}$) or translational ($\delta_{\beta p}$) mutational effect is applied in such a way that it increases or decreases expression by a fraction of its current level. A new mutant fitness $F_j$ is

obtained from the resulting cumulative protein abundance and/or transcription rates, according to the minimal model or its precision-economy version. By comparing this fitness to its ancestral value $F_i$, a fixation probability is computed [34], according to which the mutation is instantly fixed or lost at random. The mutation-selection process is repeated until a realistic level of protein abundance divergence is reached (Methods).

cumulative expression of both duplicates, such that the protein abundance of the ancestral gene is effectively subfunctionalized between the two copies—has been shown to likely be a major mechanism for the long-term maintenance of WGD-derived paralogs [23]. Under this model, as well as within our minimal framework, the individual expression levels of two duplicates can vary as long as their sum remains close enough to an optimum, resulting in a compensatory drift of expression levels as mutations accumulate [23, 24]. Quantitative subfunctionalization can of course be accompanied by functional changes within proteins, both simultaneously [42] or sequentially [43]. To ensure the generality of our minimal model, we nonetheless choose to ignore all forms of divergence other than expression changes. Functional variation is indeed much more difficult to quantify and probably occurs through a greater variety of mechanisms. Within the scope of this work, this simplification appears reasonable, as analyses show that the transcriptional bias of expression divergence is at worst weakly related to various proxies for the divergence of molecular functions among yeast paralog pairs (S5 Fig). Accordingly, we assume that the two copies comprising any paralog pair are identical and express the same protein, as could be expected early after a duplication event.

More formally, we model the evolution of the expression levels of two copies (paralogs) of the same gene across a landscape of diagonal fitness isoclines where the optimum is along a central diagonal of constant (optimal) cumulative expression (Fig 2B I). Such a landscape is obtained from a parabolic function of fitness according to the cumulative protein abundance of both paralogs. For this minimal model to be more directly applicable to the complete yeast genome, a family of functions with varying curvatures (Fig 2B II)—taken from a distribution inferred from [25]—is defined, so that distinct fitness landscapes can be obtained for different gene pairs. The model is additionally complexified in two ways to implement the precision-economy trade-off, such that this hypothesis can be tested. First, expression noise (and thus the importance of precision) is explicitly taken into account by considering the mean fitness of a population of cells expressing two paralogs at a mean cumulative protein abundance $P_{tot}$ with standard deviation $\sigma_{tot}$, which itself depends on the relative contribution of transcription to overall expression [25] (Fig 2B III and Methods). Second, economy considerations are implemented by the addition of a cost of transcription, in the form of a penalty $C$ to fitness increasing linearly with the total number of transcribed nucleotides [25] (Fig 2B IV and Methods).

To compare our two hypotheses, we performed *in silico* evolution. Only transcription and translation rates were evolved, while mRNA and protein decay rates were assumed to be constant (Methods). This restriction to only the two traits of interest had the advantage of reducing the parameter space we had to explore, while being a reasonable simplification. Most of the variation in gene expression levels indeed occurs at these two regulatory levels [25]. In addition, taking into account changes in mRNA decay has little impact on the patterns of transcriptional divergence, as we show (Fig 1A and 1B and S1 Fig). The simulations were carried out following a sequential fixation approach [44], meaning that each successive mutation was instantaneously brought to fixation or rejected.

Each simulation run was initialized from randomly generated singleton genes, each duplicated into two paralogs retaining the ancestral $\beta_m$ and $\beta_p$ rates (Methods). We assumed that a duplication event causes a doubling of total transcriptional output without affecting the

translation rate of individual transcripts, which is realistic since the translation rates of most mRNA do not change upon gene copy number increase [45]. Previous descriptions of the quantitative subfunctionalization framework, which we used as a minimal model, have postulated that the initial post-duplication cumulative protein abundance is already optimal [23, 24]. In order to be more general, we instead simply assumed that the immediate loss of a newly created paralog is deleterious. Due to the high frequency of loss-of-function mutations, any duplicate whose loss was neutral or beneficial would rapidly be lost even if it did reach fixation [46]. Accordingly, only simulated gene pairs for which the instant loss of a paralog would be deleterious were considered, while the post-duplication optimum of cumulative protein abundance was set to slightly less than double the ancestral protein abundance (1.87×)—such that the duplication-induced doubling would overshoot the optimum but still result in a positive fitness (S1 Methods).

Simulated paralog pairs were subjected to successive mutation-selection rounds, so that their transcription and translation rates could evolve (Fig 2C). During each such round, a mutation affecting the $\beta_m$ or $\beta_p$ of one of the two paralogs is attempted, and later filtered by selection according to its effect on fitness. Whether each individual mutation acts on transcription or on translation is chosen randomly, following relative probabilities $P_{\beta_m}$ and $P_{\beta_p}$ which represent the relative mutational target sizes of the two traits. Additionally, because only one gene copy is affected at a time, all mutations can be considered as *cis*-acting ones occurring in the regulatory or coding sequences of a paralog. Changes in *trans*, which affect both duplicates simultaneously and would not result in expression divergence between two fully identical gene copies, are therefore ignored. The subsequent filtering of mutants is performed in accordance with a fixation probability computed using a modified Metropolis criterion [34], which can be adjusted for different levels of selection efficacy with a parameter $N$ analogous to effective population size. This complete mutation-selection process was repeated numerous times within each simulation, until the median relative protein abundance divergence (Eq 1) for the simulated set of duplicate pairs was not significantly different from the empirical value observed for the appropriate reference set of yeast paralogs (Methods).

## The precision-economy trade-off promotes transcriptional divergence

We first performed a small-scale 'mock' simulation of 50 randomly generated duplicate pairs according to the precision-economy implementation of our minimal model. Transcription and translation were assumed to have equal mutational target sizes ($P_{\beta_m} = P_{\beta_p}$), so that only the effect of the precision-economy trade-off was tested.

This simulation revealed a clear bias towards transcriptional changes, with the relative divergence in transcription accounting for almost all the total variation of protein abundance (Fig 3A). After $\sim$ 6500 mutation-selection rounds, the median relative divergence was nearly twice larger for $\beta_m$ than for $\beta_p$ across the 50 simulated paralog pairs. The resulting evolutionary trajectories highlight that expression divergence was driven by transcriptional changes, as the most transcribed paralog is almost always associated with the highest protein abundance, while the same is not true for the most translated one (Fig 3C). Interestingly, these patterns do not arise because the divergence of transcription levels is itself beneficial under the precision-economy trade-off, as illustrated by the fitness plateau observed from round 1000, while transcription rates are still actively diverging (Fig 3B). Indeed, since expression noise scales according to both protein abundance and transcription rate (Methods), variance in the cumulative expression of two paralogs depends on their total $\beta_m$ and expression. Similarly, the cost of transcription only depends on the total number of mRNAs synthesized. As such, both the precision and economy of expression are not impacted by the distribution of the relative

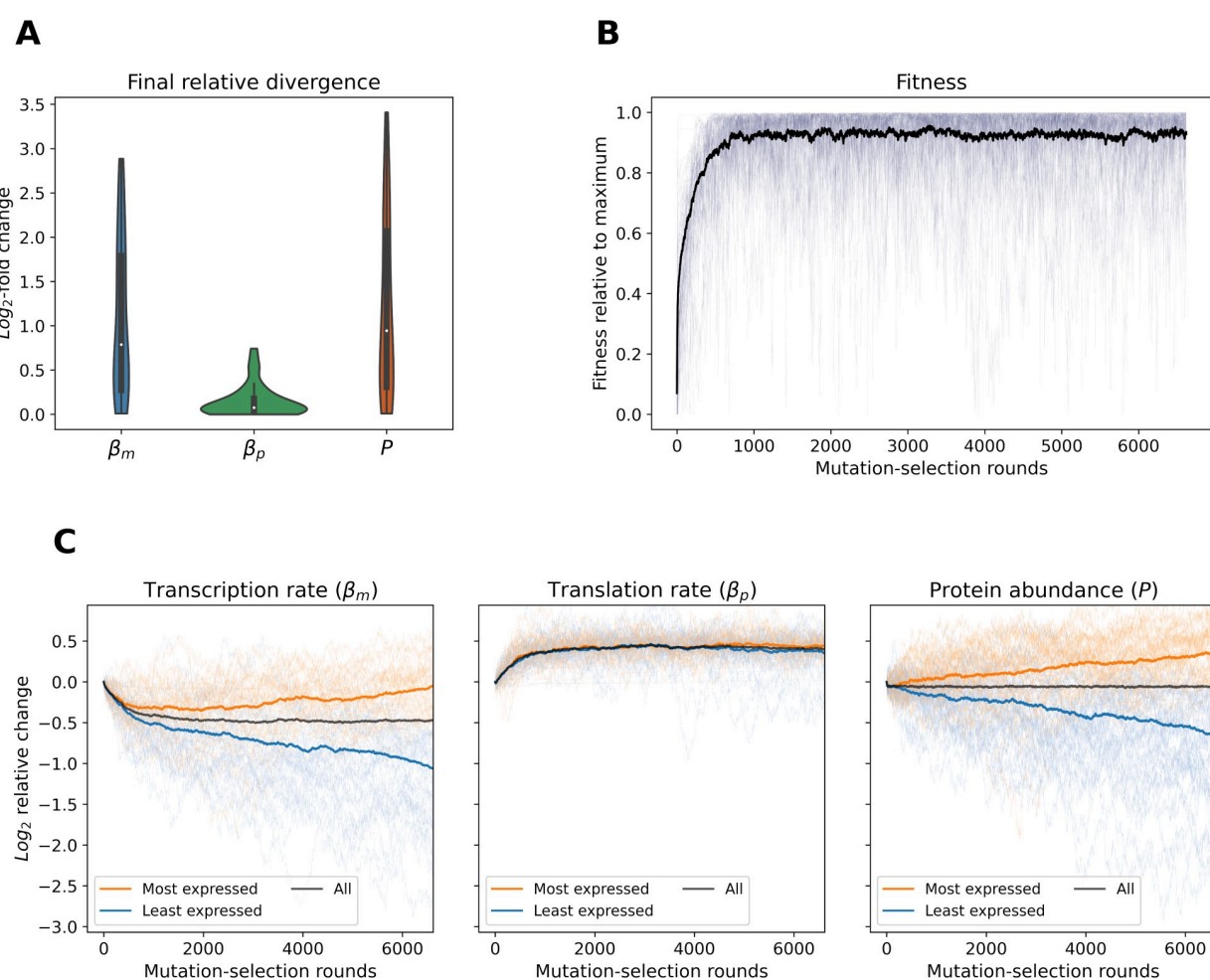

**Fig 3. The precision-economy trade-off favors transcriptional divergence.** (A) Distributions of the final log-fold relative divergence in transcription, translation and protein abundance for the simulation of 50 randomly generated paralog pairs. The standard deviation of mutational effects was set to an arbitrarily small magnitude ($\sigma_{mut}$ = 0.025) and a scenario of high selection efficacy was considered ($N = 10^6$). The simulation was stopped once protein abundance divergence matched that observed in yeast WGD-derived duplicates (Methods). (B) Fitness for each of the 50 duplicate pairs (faint lines) throughout the simulation. The mean value at any time point is shown by the black line. Fitness is scaled individually for each gene pair between its minimum (0) and maximum (1) values reached during the simulation. (C) Full evolutionary trajectories of the 50 simulated paralog pairs in transcription rate, translation rate and protein abundance. All three features are shown as log-scaled relative changes from the ancestral value for each duplicate pair and the darker lines represent the means through time.

transcriptional contributions between two paralogs, which thus has no effect on fitness. Rather, it is the ratio of the cumulative translation and transcription rates which dictates fitness under precision-economy constraints, as clearly shown by the early adaptation phase of the simulated evolutionary trajectories (Fig 3B). After an initial reduction of cumulative protein abundance to reach the new post-duplication expression optimum, the transcription of both duplicates is decreased while their translation is increased (Fig 3C), to rebalance the $\frac{\beta_p}{\beta_m}$ ratio [25] of the paralog pair. Following this first phase of post-duplication adaptation, compensatory drift of cumulative protein abundance takes place [23, 24], but changes are almost exclusively biased towards transcriptional divergence. This occurs due to interactions between transcription- and translation-acting mutations introduced by the precision-economy trade-off. Further transcriptional changes can for instance be expected to be favored by selection

after the fixation of a mutation altering $\beta_m$, as they have the potential to compensate effects on both precision and economy, while a change of translation can individually only act on precision.

This first simulation thus shows that the trade-off between the precision and economy of gene expression can lead to a mostly transcriptional expression divergence between two paralogs, potentially creating evolutionary patterns as observed in yeast.

## Selecting a biologically plausible parameter space

The precision-economy trade-off is sufficient to favor the transcriptional divergence of duplicated genes, but it might still not be the most likely explanation for the evolutionary patterns observed in yeast paralogs. To rigorously compare our two competing hypotheses, we performed series of large-scale simulations under biologically realistic assumptions. All the corresponding *in silico* evolution experiments were carried out in three replicates of 2500 randomly generated paralog pairs. Throughout these computational experiments, the WGD-derived duplicates of *S. cerevisiae* were used as a reference set, both for identifying when to stop iterating through mutation-selection rounds as well as for assessing the results of each simulation. We chose to restrict our analysis to this group of paralogs, because the mechanism of quantitative subfunctionalization, which we use as a minimal evolutionary model, was initially proposed in the context of whole-genome duplication [23] and its applicability to other duplication events is thus less clear. An additional reason to focus on WGD-derived paralogs is that our initial observation is less certain for duplicates which originated from SSD. In one instance, controlling for variations in mRNA decay erased the difference between the magnitudes of transcriptional and translational divergence within this set of gene pairs (Fig 1B). Ensuring the biological plausibility of our simulations also required a careful examination of two important parameters—the efficacy of selection $N$ and the standard deviation of mutational effects $\sigma_{mut}$ –, for which there are only partial estimations.

Many estimates of the effective population size of *Saccharomyces* yeasts, which is analogous to the $N$ parameter of the evolutionary algorithm regarding the efficiency of selection [34], are available [32, 33]. The relevance of such historical values to the immediate post-WGD context is however unclear, because the whole-genome doubling could have been accompanied with a strong founder event. Identifying a most realistic $N$ is therefore difficult. Accordingly, we instead chose to consider two scenarios, respectively of high ($N = 10^6$) and reduced ($N = 10^5$) efficacy of selection, and compare our two hypotheses in both contexts.

Similarly, only limited information is available on the effect of mutations—and especially *cis*-occurring ones—on expression. Previous studies often focused on only a small subset of genes [47, 48], did not differentiate between *cis* and *trans* mutations [48, 49], assessed too few mutations [49] or were limited to substitutions occurring within a short segment of the promoter sequence [50]. Thus, instead of arbitrarily choosing a $\sigma_{mut}$, we performed simulations across a range of standard deviations to identify the most biologically plausible value. As most mutations affect the expression level of selected yeast genes by 20% or less [48], we considered values ranging from 0.01 to 0.35 (1% to 35%).

Our second hypothesis only stipulates that the regulation of transcription may have a larger mutational target size than that of translation without specifying the magnitude of this difference. Testing it therefore requires using various relative probabilities of transcriptional and translational mutations, $P_{\beta_m}$ and $P_{\beta_p}$ (Fig 2C). To ensure the robustness of the identification of the best-fitting $\sigma_{mut}$, it was combined with this screening into a grid search, which was performed separately for the two scenarios of selection efficacy (Methods). This approach identified best $\sigma_{mut}$ of 0.025 and 0.075, respectively under high ($N = 10^6$) and reduced ($N = 10^5$)

selection efficacies (S6 Fig). Interestingly, the value obtained for $N = 10^6$ is highly consistent with a previous experimental characterization of *cis*-regulatory mutations in the yeast *TDH3* promoter [47]. Whether the latter is representative of the typical *S. cerevisiae* gene is however unclear [48].

## A difference of mutational target sizes may better explain the observed divergence patterns

We next assessed the extent to which our minimal model and its precision-economy version could replicate the main features of the divergence patterns of yeast paralogs (Fig 1): 1) a pre-dominance of transcriptional changes, 2) a weak positive correlation between the magnitudes of relative divergence in transcription and in translation and 3) a high frequency of amplifying changes at the two regulatory levels.

We focused on simulations performed in three replicates of 2500 paralog pairs using the best-fitting $\sigma_{mut}$ values identified previously, for the two selection efficacy regimes ($N = 10^6$ and $N = 10^5$). At the end of each simulation, summary statistics were computed on the set of 2500 diverged gene pairs to test whether each of the three empirically observed properties could be replicated (Fig 4A).

We first used a mean Kolmogorov-Smirnov (KS) statistic—comprised between 0 and 1 and for which a lower value indicates a better fit—to quantify the overall distance between the simulated and empirical distributions of relative divergence ($log_2$-fold change) in transcription and translation rates as well as in protein abundance (Fig 4A). This comparison reveals the contrasting performance of the two models, depending on the choice of parameters (Fig 4B).

When a high efficacy of selection is assumed, the minimal model is by far the most accurate, as shown by the attainment of much lower mean KS statistics (as low as $\sim 0.07$, compared to values $> 0.2$ for the other model). The best fit is obtained when a higher probability of mutations affecting $\beta_m$ is assumed, and especially when $P_{\beta m}/P_{\beta p}$ is between 3 and 6, which supports the hypothesis of a larger mutational target size for transcription. The poor performance of the precision-economy model in these conditions is almost entirely due to its inability to produce a realistic translational divergence (S7A and S7B Fig). We thus performed additional simulations with $P_{\beta p} > P_{\beta m}$, which however did not result in a better fit (S8A and S8B Fig). In contrast, when the efficacy of selection is reduced, the performance of the two models is much more similar, as illustrated by the obtention of mean KS statistics below 0.1 in both cases (Fig 4B). The most accurate replication of the empirical relative divergence distribution is still observed for the minimal model, more precisely when transcriptional mutations are three to six times more likely than translational ones, but the precision-economy trade-off performs almost as well when mutations affecting $\beta_p$ are as frequent or even twice likelier than ones on $\beta_m$. Supplemental simulations show that increasing the relative probability that mutations act on translation rates—up to $P_{\beta m}/P_{\beta p} = 1/10$—does not further improve the performance of the precision-economy model (S8C Fig). This overall highlights how both hypothesized mechanisms might have shaped the expression divergence of yeast paralogs, although the mutational target size hypothesis appears much more robust to assumptions about the values of evolutionary parameters. Which explanation is most likely to apply will ultimately depend on empirical measurements of parameter values that we can at best approximate here.

We also computed the two types of divergence correlations used previously—either between the transcription and translation $log_2$-fold changes or between their signed versions—on the complete set of 2500 diverged paralog pairs obtained from each replicate simulation (Fig 4A).

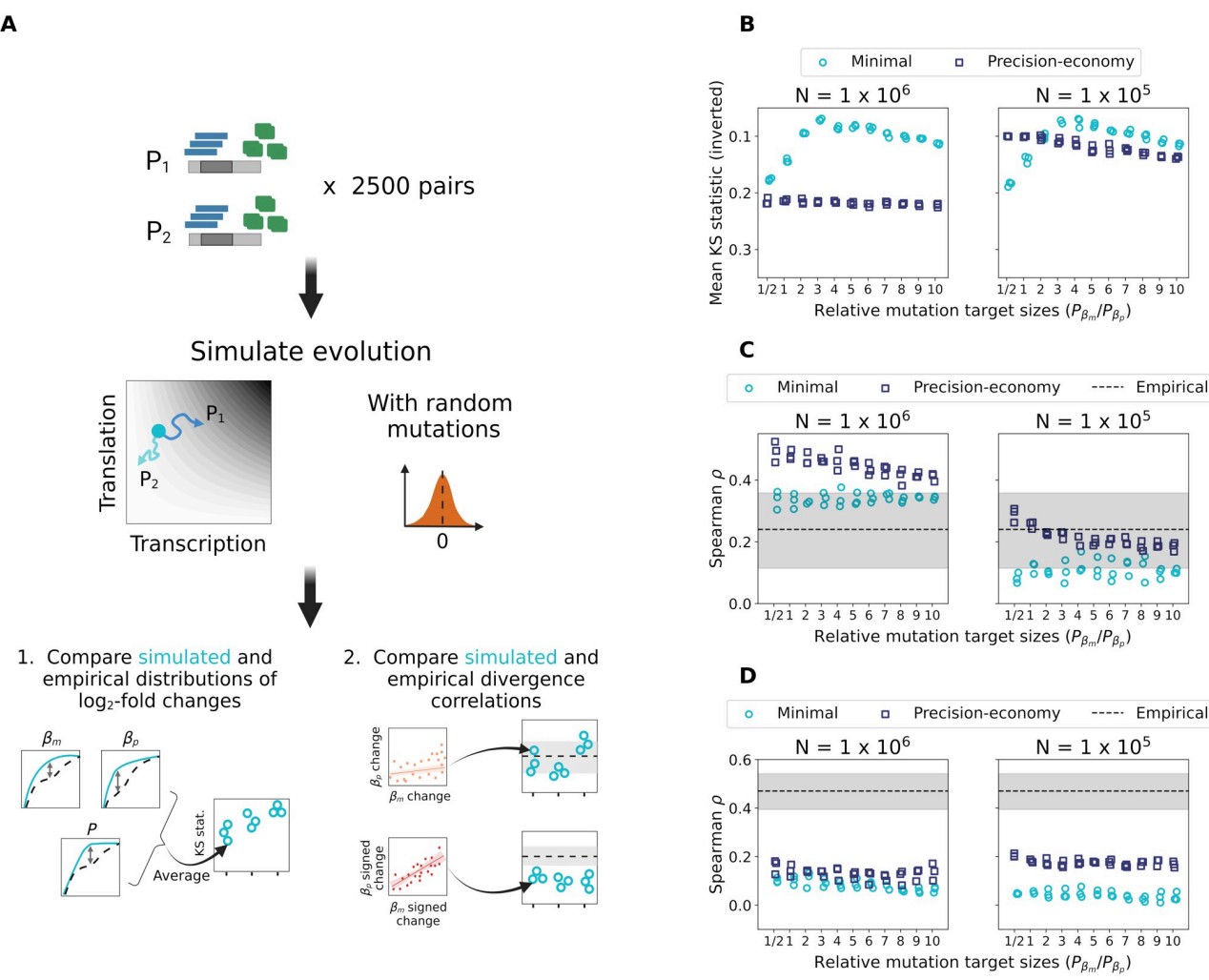

**Fig 4. A larger mutational target size for transcription more robustly replicates the expression divergence patterns of yeast WGD-derived paralogs across levels of selection efficacy.** (A) For each combination of model and parameters, three replicate simulations of 2500 paralog pairs were performed. Two types of summary statistics were computed to compare simulation results to empirical observations. The Kolmogorov-Smirnov (KS) statistic, equal to the largest difference between two cumulative distribution functions, was used to quantify the distance between simulated and empirical distributions of $log_2$-fold changes in transcription, translation and protein abundance (top). For each replicate simulation, the three resulting KS statistics were combined into a single mean value. The two divergence correlations between transcriptional and translational changes were also calculated on the set of paralog pairs obtained from each simulation, resulting in three measurements for each combination of model and parameter values (bottom). Created with BioRender.com. (B) Replication of the distributions of relative divergence in transcription rate, translation rate and protein abundance following simulations, as shown by the mean KS statistics obtained from the pairwise comparisons of all the corresponding empirical and simulated distributions. (C) Correlations between the magnitudes of transcriptional and translational $log_2$-fold changes for gene pairs evolved *in silico* according to the minimal and precision-economy models. The shaded area shows the 95% confidence interval for the empirical correlation observed in *S. cerevisiae* WGD-derived paralogs. (D) Correlations between signed $log_2$-fold changes in transcription and translation rates for the same simulated gene pairs. The shaded area represents the 95% confidence interval on the empirical correlation.

The replication of the first correlation, which reflects the positive association between the absolute magnitudes of transcription and translation changes (Fig 1C), is again dependent on the choice of model and selection efficacy regime, but much less on the relative frequencies of transcriptional and translational mutations. When $N = 10^6$, it is reasonably reproduced only by the minimal model, for which almost all simulations result in a correlation that falls within the 95% confidence interval of the empirical value (Fig 4C). For $N = 10^5$, implying a reduced efficacy of selection, it is instead the precision-economy trade-off which is associated with the

most realistic such correlations. All the corresponding simulation runs indeed produce correlation coefficients within the confidence interval. The minimal model is however also able to generate realistic correlations, for one or two replicates under most of the tested ratios of transcriptional and translational mutational target sizes (Fig 4C). This again highlights the plausibility of the two alternative hypotheses. Yet, it also further reveals the increased robustness of the minimal model.

The second correlation, showing that transcriptional and translational changes often occurred in the same direction (Fig 1D), could be replicated by neither model, across the selection efficacy regimes as well as the range of mutational target size ratios tested (Fig 4D). Nevertheless, all simulations are interestingly associated with strictly positive correlations, even the minimal model, contrary to the intuitive expectation of a compensatory drift at both levels to maintain protein abundance [23, 24]. This, as well as the inability to replicate the empirical correlation, may reveal an effect of the postulated post-duplication expression optimum. We thus performed additional simulations while varying this optimum, to consider the alternative possibilities that the duplication-induced expression doubling was perfectly optimal or did not sufficiently increase the expression level. These showed that the correlation between signed relative divergences is mostly independent from the posited expression optimum (S9B Fig), ensuring that we were not being misled by a potentially poor choice of this value. We note that the effect of the post-duplication optimum is stronger for the correlation between the absolute magnitudes of the $log_2$-fold changes (Fig 4C and S9A Fig), but still insufficient to invalidate our previous conclusion on the replication of this feature.

Overall, these comparative analyses show that both mechanisms may realistically have contributed to the greater importance of transcriptional changes in the expression divergence of yeast WGD-derived paralogs. They however also reveal that the two hypotheses cannot fully explain the observed evolutionary patterns—since one of the three features of the empirical divergence can be replicated by none of the models—and as such require refining. The hypothesis of a larger mutational target size for transcription emerges as the preferred explanation, since it results in the best fit (the lowest mean KS statistic found on Fig 4B) while being more robust to assumptions about the efficacy of natural selection. Although the relative mutational target sizes of transcription and translation regulation are not known, the fact that the best agreement with our observations is obtained for a modest difference of relative mutation probability means that the bias need not be important to impact evolution.

## Revisiting the hypotheses when considering transcription-translation couplings and biased mutational effects distributions

Some of the assumptions we made about the mutational process may be unrealistically simple, and this might explain both models' inability to replicate the strong positive correlation observed between signed transcriptional and translational changes (Fig 1D). First, the extent to which mutations may independently act on transcription and translation is unclear. Second, mutational effects might not be distributed symmetrically.

Many mutations in the transcribed region of a gene may for instance simultaneously have transcriptional and translational effects, as the identity of the translated codons might affect both mRNA stability and translation itself [51–53]. In addition, mutations at one regulatory level may be associated with amplifying or buffering regulatory changes at the other—as suggested by stress responses [54–56]. The effects of random mutations on expression level might also often be distributed asymmetrically, as shown by the experimental characterization of mutations affecting ten yeast genes [48]. Recent work additionally predicted that mutations

increasing expression are rarer for highly expressed promoters, and vice-versa for lowly expressed ones [50]. As such, there are many potential constraints on the effects of mutations which could create correlations between transcriptional and translational changes. Taking these effects into account may allow at least one of our models to fully replicate the expression divergence patterns of yeast WGD-derived paralogs.

While it is not possible to include all of the complexity of gene regulation in a single model, we examined these additional potential factors. We made two new versions of our simulation framework, respectively implementing an asymmetry in the distribution of mutational effects and a correlation between the transcriptional and translational effects of mutations. This second addition to the model, which allows mutations to act on both $\beta_m$ and $\beta_p$ at once, could account for regulatory responses as well as for the potential coupling between mRNA stability and translation efficiency. Within our minimal model, under which identical changes to transcription and mRNA stability are entirely equivalent (as only their effect on protein abundance matters), correlated effects on $\beta_m$ and $\beta_p$ can indeed represent such a coupling.

Mutation asymmetry was implemented using a skew normal distribution of mutational effects with skewness parameter $\alpha \neq 0$, while correlations between transcriptional and translational mutations were added using a bivariate normal distribution of mutational effects. Because the latter modification meant that each mutation now affected both transcription and translation, differences of mutational target size between the two traits had to be modeled differently, using the effect size of mutations (Methods). An additional grid search was also required to identify the best-fitting standard deviations of mutational effects to use in subsequent simulations (S10 Fig).

Simulations were performed as previously across ranges of distribution asymmetry and correlation of mutational effects (Methods). While both negative and positive correlations between transcriptional and translational effects were assayed, only negative skewness values —biasing mutations towards a decrease of expression—were used, since a positive skew lengthened simulations too much by impeding the initial protein abundance reduction. The accuracy of each simulation run was again assessed using summary statistics computed on the complete set of 2500 paralog pairs simulated (Fig 5A). For all three metrics used previously, the values obtained for each of three replicate simulations were combined into a mean —or grand mean in the case of KS statistics –, which was then used in comparisons of model and parameter values combinations. An additional set of summary statistics was also computed: p-values of Mood's median test for the comparison of empirical and simulated distributions of relative divergence at the levels of transcription, translation and protein abundance. These allowed to classify each simulation run as generating expression divergence of a realistic magnitude or not. When considering our minimal model of post-duplication evolution in a context of high selection efficacy ($N = 10^6$), both tested mutational constraints are sufficient to create a strong positive correlation between transcriptional and translational signed $log_2$-fold changes, as observed within real WGD-derived paralog pairs (Fig 5D). A high negative skew on distributions of mutational effects or a strong positive correlation between effects on $\beta_m$ and $\beta_p$ can both result in Spearman's correlation coefficients which fall within the empirical confidence interval, for a wide range of mutational target sizes ratios. This can even coincide with the obtention of realistic relative divergence distributions, as shown by non significant Mood's median tests ($p > 0.05$) on all three properties and low grand mean KS statistics (Fig 5B). There is however no combination of parameters for which the other divergence correlation—between the absolute magnitudes of fold changes—can simultaneously be replicated (Fig 5C). As such, the addition of more realistic mutational constraints can rescue one type of divergence correlation at the expense of the

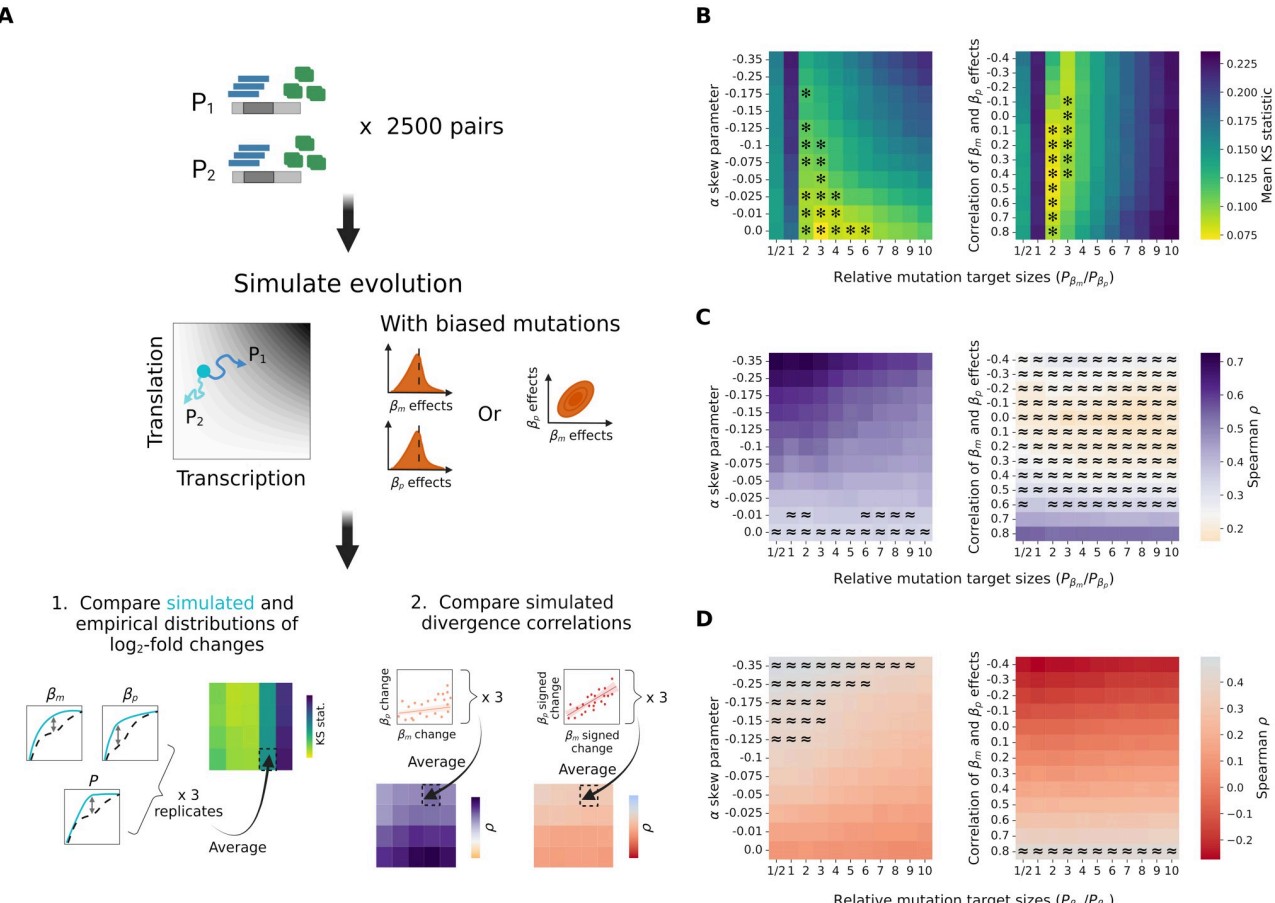

**Fig 5. Adding skewness and transcription-translation correlations to the distribution of mutational effects affects the replication of the divergence correlations.** Results shown for evolutionary simulations performed under the minimal model with the assumption of $N = 10^6$. (A) For each combination of parameters, three replicate simulations of 2500 paralog pairs were performed, and summary statistics were computed as previously. The mean KS statistics (top) and divergence correlations (bottom) obtained for each replicate were combined into a grand mean KS statistic and a mean correlation coefficient, respectively. Created with BioRender.com. (B) Replication of the three distributions of relative divergence (transcription, translation and protein abundance) for a range of mutational effects distribution asymmetry (left) or correlations between the effects of transcriptional and translational mutations (right), as shown by grand mean KS statistics. Asterisks identify instances where all three magnitudes of relative divergence are realistic for at least one of three replicate simulations ($p > 0.05$, Mood's median test). (C, D) Average final correlation across the three replicate simulations between 1) the magnitudes of transcriptional and translational $log_2$-fold changes (in C) and 2) the signed $log_2$-fold changes in transcription and translation (in D) across the same ranges of mutational effects distribution asymmetry or correlations between the effects of transcriptional and translational mutations. In each case, ≈ designates parameter combinations where a correlation coefficient within the 95% confidence interval of the empirical value was obtained for at least one replicate simulation.

other, which was in contrast properly replicated by our previous simulations assuming perfectly independent mutations and normally distributed mutational effects. An identical conclusion is reached when the efficacy of selection is reduced (S11 Fig) or when the precision-economy model is instead considered (S12 Fig).

While this attempt at more realistically modeling the effect of mutations did not clearly favor one hypothesis or the other, the fact that it rescued the replication of one feature of the expression divergence of yeast paralogs (Fig 1D) suggests that at least one of our two models may be adequate when combined with truly realistic mutational biases and correlations. A larger mutational target size for transcription and the evolutionary trade-off between the precision and economy of gene expression thus both appear as suitable non-mutually exclusive

explanations for the predominance of transcriptional changes in the divergence of yeast paralogs.

## Discussion

Our analysis of published data [25] suggests that transcriptional changes played a greater role than translational ones in the divergence of paralogs in the yeast *S. cerevisiae*. Whether this is a general feature of the evolution of duplicated genes remains to be fully investigated, but a report of very similar evolutionary patterns in plants *A. thaliana* and *Z. mays* [15] highlights the plausibility of such a generalization.

Focusing more specifically on WGD-derived paralogs, we used *in silico* evolution to investigate two potential mechanisms explaining this predominantly transcriptional divergence: an evolutionary trade-off between the precision and economy of gene expression [25], and a larger mutational target size for the regulation of transcription. Simulations revealed that both hypotheses may be consistent with the observed patterns of evolution. We turned to WGD-derived paralogs because the minimal model of post-duplication evolution we used was initially described for cases of whole-genome doubling [23]. It might however still be applicable to other types of duplication events. Our framework for instance also partially replicated the divergence patterns of SSD-derived paralog pairs, albeit with a lower overall agreement (S13 Fig).

The precision-economy trade-off is sufficient for divergence to occur mostly in transcription, which highlights how interactions at two levels of expression regulation can shape evolutionary trajectories. The general relevance of this type of epistasis in the evolution of duplicate genes is however unclear. Due to the small magnitude of the fitness effects involved [25], precision-economy constraints may only be impactful in large populations where selection is particularly efficient. This might not be the case in the early evolution of WGD-derived paralogs, even in *S. cerevisiae*, as WGD events initially create a small polyploid population. We also note that, contrary to what the $N$ parameter of $10^5$–$10^6$ would suggest, our simulations are not exactly representative of such a large finite population. Because any beneficial or perfectly neutral mutation is automatically accepted [34], even infinitesimal fitness gains are visible to selection, rather than only those larger than $1/N$. In a more realistic scenario, weakly beneficial and mildly deleterious mutations acting on transcription or translation may both be close to neutrality and have similar fixation probabilities, limiting the ability of the precision-economy trade-off to favor transcriptional divergence.

A larger mutational target size for transcription, modeled as a higher relative mutation probability, emerges as the preferred hypothesis, as it is robust to assumptions about selection efficacy. The high similarity between the expression divergence patterns of WGD- and SSD-derived paralogs (Fig 1 and S1 Fig; [15]), which have been shown to differ substantially in their initial properties and subsequent evolutionary trajectories [57–59], may also support such a more general mutational mechanism. In addition, when strictly considering *cis* mutations—the only changes which can cause two identical gene copies to diverge—as in the current work, a larger mutational target for transcription is intuitively likely. Because *cis* mutations acting on translation have to occur within the transcribed sequence while transcriptional mutations can also arise upstream or downstream of the gene, more nucleotide positions could potentially affect transcription. Determining whether a higher frequency of mutations affecting transcription truly contributes to the predominance of transcriptional changes within paralog pairs would however require direct measurements of the neutral evolutionary rates of transcriptional and translational efficiencies.

Since the current distribution of transcription and translation rates among *S. cerevisiae* genes has previously been attributed to the optimization of the precision-economy trade-off [25], suggesting that such constraints may not be needed to explain the divergence patterns of paralogs might appear contradictory. This is especially true considering the significant energetic costs of even small increases of transcription and translation [37, 60]. Yet, even when precision-economy considerations are fully neglected under the minimal model, extended *in silico* evolution results in only minor deviations from the reported distribution of genes in the transcription-translation space (S14 Fig). One plausible explanation could be that precision-economy constraints impact evolutionary trajectories on longer timescales and/or along greater ranges of variation, while mutational effects dominate on the shorter timescales and smaller expression changes associated to the divergence of duplicated genes.

While we have identified two potential underlying mechanisms for the preeminent contribution of transcriptional variation to the expression divergence of paralogs, their wider applicability to the evolution of gene expression levels remains to be investigated. Whether any of them could explain general trends of faster transcriptional evolution within and between species [6–8] is indeed unclear. If a larger mutational target size for transcription was involved, it would presumably affect singleton genes as well as paralog pairs—unless it were unique to duplicates, potentially because more nucleotide positions might affect transcription than translation in *cis*. In contrast, if the precision-economy trade-off was responsible for the greater magnitude of transcriptional divergence among yeast paralogs, it likely would not explain any general tendency for transcription to evolve at a faster rate. It is indeed interactions between mutations in the two paralogs that favor transcription divergence and bias expression changes within our simulation, which could hardly apply to singletons genes. The precision-economy trade-off may nevertheless have less intuitive effects on the evolution of expression levels in such genes if combined with other mechanisms, for instance a slight difference of mutational target sizes. How this trade-off [25] could affect the relative evolutionary rates of transcription and translation at the genome scale, particularly over long timescales, thereby warrants further investigation.

Besides providing explanations for the patterns of expression divergence observed in yeast paralogs—as well as suggesting hypotheses for a more general propensity for transcription changes to dictate protein abundance variations –, we also illustrate how biases in the mutation process can impact the multi-level evolution of expression levels. Asymmetry in the distribution of mutational effects and correlations between transcriptional and translational effects both markedly affect the correlation between transcription and translation changes within gene pairs in our simulations. This further shows that various types of mutational bias could impact the evolutionary trajectories of a duplicate pair under selection to maintain its cumulative protein abundance, as might be the case for singleton genes in the absence of selection [48]. Overall, our work thus highlights the importance of thoroughly characterizing the distributions of mutational effects on expression at multiple regulatory levels in order to fully understand the expression divergence of paralogs, and, more widely, the genome-wide evolution of expression levels. Future experiments could for instance perform large-scale measurements of the transcriptional and translational effects of mutations and assess what correlation(s) exist between them.

Irrespective of its cause(s), the predominance of transcriptional changes in the expression divergence of paralogs, whether yeast-specific or more general, might have significant evolutionary consequences. Due to the relationship between transcription rate and expression noise [25, 35, 36], it could result in a greater divergence of noise levels within duplicate pairs than symmetrical or mostly translational expression changes. Whether this could affect or even dictate the evolutionary paths followed by paralogs remains to be investigated. One intriguing

possibility is that such early divergence of noise levels could favor the resolution of noise-control conflicts [61] while maintaining functional redundancy. This is exemplified by paralogous yeast transcription factors *MSN2/MSN4*, which appear to combine the benefits of low and high expression noise. While one gene is stably expressed with low noise across conditions, the other is expressed with high noise and is environmentally responsive [62]. Although the maintenance of this gene pair likely involved other mechanisms such as post-translational changes [63], it is conceivable that the benefit of such two-factor regulation was first revealed by early transcriptional divergence, and later refined through changes in promoter architecture. A high prevalence of such trajectories could help explain why 40% of WGD-derived transcription factor pairs in *S. cerevisiae* bind the same targets [64]. While speculative, this possibility underscores how a simple bias towards transcriptional divergence may have far-reaching impact.

We studied the expression divergence of paralogs while neglecting all types of changes to protein function and regulation. This simplification is supported by the absence of a clear continuous relationship between functional divergence and the relative contribution of transcriptional variation to expression changes, although stronger relationships with the magnitude of expression divergence advise caution (S5 Fig). Such an approach would be particularly suitable if expression changes were not affected by functional divergence, which could for instance be true if expression diverged first during evolution. This was postulated by previous models, according to which early expression changes under selection to maintain cumulative protein abundance lengthen the retention of duplicates and allow function-altering mutations to arise and fix [23, 43]. Whether the divergence of protein function really follows expression changes is however unclear, as it might also occur simultaneously or even precede it [42].

If expression divergence did occur first in the evolution of duplicated genes, it would also shape their subsequent functional divergence. Any mutation affecting the function of one of two identical paralogs needs to overcome the deleteriousness of the associated reduction in the abundance of the ancestral gene product [23], meaning that such changes might be restricted to new functions for which the protein abundance of the mutated gene copy is already close to optimal. The addition of precision-economy constraints suggests an intriguing extension of this model. If transcription cost and tolerance to expression noise dictate optimal transcription and translation rates for all genes as described by [25], functional changes may be further restricted to molecular functions which are compatible with the precision and cost of the expression of the affected paralog.

While this work provides insights on the expression divergence of duplicated genes—and thus more generally on the evolution of gene expression –, it presents limitations. A first one is the assumption that all paralog pairs were initially made of two identical gene copies. This is especially significant in the case of WGD-derived pairs, as the yeast WGD likely involved a hybridization event [65]. As such, some paralogs were already diverged and their expression fold changes may not be representative of the evolution of duplicates. The agreement between WGD- and SSD-derived pairs would however support the idea that a predominance of transcriptional changes is a general feature of the divergence of paralogs. Other simplifying assumptions made to simulate the evolution of *S. cerevisiae* WGD-derived duplicated genes also warrant examination. Sequential fixation, where mutants never coexist in the population [44], is likely not fully representative of evolutionary processes in yeasts with large population sizes. It is also unable to replicate the fixation dynamics of pairs of compensatory mutations [66], which could play a role in the expression divergence of duplicated genes. More importantly, the use of the Metropolis criterion to accelerate simulations may skew the resulting evolution patterns by equally valuing all positive fitness changes and artificially widening the gap between mildly beneficial and slightly deleterious mutations. Another important limitation of our approach is the absence of gene loss. Post-WGD paralog retention is known to be on the

order of 15% in *Saccharomyces* yeasts after $\sim$100 million years of evolution [67], yet all randomly generated duplicate pairs are retained throughout our simulations. Including loss-of-function mutations, through which gene copies could have been inactivated when tolerated by selection, would have been more realistic, but tests showed that it made the end condition of the simulation an ever-moving target. We note that performing all tolerated loss-of-function mutations at the end of the simulations (Methods), prior to the calculation of summary statistics, produces qualitatively similar results (S15 Fig), which strengthens our conclusions. An alternative could have been to restrict our simulations to duplicate pairs which were destined to be retained for an extensive period. Using the current transcription and translation rates of paralogs to infer the expression levels of ancestral singletons and then investigate the divergence of duplicates would however have proved circular. Caution is additionally in order before generalizing our observations, and the two underlying mechanisms investigated, to the evolution of all duplicated genes. The patterns of divergence among paralogs that motivated our work are by definition observed for pairs that survived to this day, such that we cannot infer what happened for pairs which returned to single-copy genes.

## Conclusion

The expression divergence of yeast paralogs mostly occurred at the transcriptional level, which may be due to two mechanisms: an evolutionary trade-off between the precision and economy of gene expression and a larger mutational target size for transcription than for translation. Whether these explanations also hold for more general observations that transcription may evolve at a faster rate within and between species remains to be elucidated. Interestingly, some features of the divergence of duplicate pairs can be replicated by either model only when mutational biases—asymmetry and transcription-translation correlations in the distributions of mutational effects—are added. This observation illustrates the importance of fully characterizing how mutations jointly affect the different traits contributing to gene expression. More importantly, our work highlights how measuring the neutral evolutionary rates of transcription and translation efficiencies is essential to a complete understanding of the evolution of expression levels. Such measurements would be pivotal in discriminating between the two alternative mechanisms, as well as in elucidating whether our findings apply only to duplicated genes. Further research will additionally help clarify the wider evolutionary implications of predominantly transcriptional expression divergence, especially regarding the impact of gene duplication events.

## Methods

### Expression divergence of yeast paralogs

**Transcriptional and translational divergence.** We downloaded the transcription and translation rates of 4440 yeast genes from [25]. These rates have been inferred from a mRNA-seq and ribosome profiling experiment by [30]. Transcription rates $\beta_m$ for each gene $i$ were obtained as follows (Eq 2). The fraction $r_i$ of mRNA-seq RPKMs $r_j$ associated to gene $i$ was first converted into mRNA abundance $m_i$ using the total number $N_m$ of transcripts per cell, estimated to 60 000 molecules. Then, $m$ was inferred from the mRNA abundance using decay rate $\alpha_{m,i}$, in hours$^{-1}$. The rates reported by [25] were obtained under the assumption that mRNA decay rate does not vary between genes, such that $\alpha_{m,i}$ becomes constant $\alpha_m$, equal to

the median transcript decay rate of 5.10 h$^{-1}$.

$$m_i = N_m \frac{r_i}{\sum_j r_j}$$

$$\beta_{m,i} = m_i \alpha_{m,i} \tag{2}$$

To obtain the translation rate $\beta_p$ for each gene $i$, the total translational flux (in proteins per h) was first estimated as the product of median protein decay rate $\alpha_p$, equal to 1.34 h$^{-1}$, and total number of proteins per cell $N_p$, amounting to $1.1 \times 10^8$ molecules [25]. The fraction of this synthesis flux directed to gene product $i$ was then obtained from the corresponding fraction $s_i$ of ribosome profiling RPKMs $s_j$, which was divided by the abundance $m_i$ of the transcript [25].

$$\beta_{p,i} = \frac{N_p \alpha_p}{m_i} \frac{s_i}{\sum_j s_j} \tag{3}$$

We identified paralogs originating from either whole-genome duplications (WGD) or small-scale duplication events (SSD) using annotations by [68]. Groups of more than two duplicates, derived from successive duplication events, were excluded by only considering genes annotated "WGD" and "SSD" in the cited work. We obtained a final set of 409 high-confidence paralog couples, of which 245 are WGD-derived and 164 have originated from SSD events.

To assess the relative expression divergence of paralogs, we computed $log_2$-fold changes in transcription rates and translation rates within each gene pair, according to Eq 1. The two correlations between transcriptional and translational divergences within paralog pairs were calculated as Spearman's $\rho$. For the correlation computed from signed $log_2$-fold changes, the latter were calculated similarly to Eq 1, but without defining the ratio as max/min. For each pair, the ratio was computed in the two possible orientations and a duplicated dataset was generated, on which the correlation was then calculated.

The 95% confidence intervals for each of the two correlations were obtained by bootstrapping, involving 10 000 sampling with replacement of 409 true pairs of observations (or 818 when using a duplicated dataset). The corresponding correlation was recomputed on each of the resampled datasets.

**Taking into account gene-to-gene variation in mRNA decay.** We recalculated $\beta_m$ using experimental measurements of mRNA decay. The $\alpha_m$ constant was replaced with gene-specific decay rates, taken from four datasets [26–29]. All measurements were converted into hours$^{-1}$. As decay rates compound active degradation and dilution due to cell division within the framework of [25], the effect of this dilution was added when necessary [26, 28, 29]. As [25], we assumed a cell division time of 99 minutes, such that the decay rate of transcript $i$ is obtained from the experimental decay constant $\gamma_i$ as follows: $\alpha_{m,i} = \gamma_i + \frac{\log(2)}{99/60}$. Relative divergences ($log_2$-fold changes) as well as the two correlations were recalculated with these datasets.

**Validating the translation rates.** For the 818 paralogs previously identified, protein abundances were computed from transcription and translation rates reported by [25] using Eq 5. To better isolate whether the translation rates $\beta_p$ are representative of translational flux, constant $\alpha_p$ was replaced by gene-specific protein decay rates $\alpha_{p,i}$ obtained from experiments [69, 70]. As for transcript decay, the effect of dilution due to cell division was added when necessary. Variations in mRNA decay were not taken into account, simply because they could not have any effect on the calculation. Any constant or gene-specific transcript decay rate would indeed be present both at the numerator—to obtain transcription rate $\beta_{m,i}$ (since $\beta_{m,i} = m_i \alpha_{m,i}$)

—and denominator. Only mRNA abundance $m_i$ for each gene $i$ (read counts obtained in mRNA-seq normalized into a number of transcripts per cell [25]) was thereby needed as transcript-level data. The abundance of each protein $i$ was thus estimated as $p_i = \frac{m_i \beta_{p,i}}{\alpha_{p,i}}$. Within each of the 409 duplicate pairs, an estimated $log_2$-fold change of protein abundance was computed from these values $p_i$. Measurements of the abundance of each protein [69, 71, 72] were also used to calculate an experimental $log_2$-fold change of protein abundance within each paralog pair. For each combination of datasets, the Pearson correlation between the estimated and experimental $log_2$-fold changes was computed. To assess the significance of the resulting correlations, we compared them to distributions of correlation coefficient obtained from 10 000 repetitions of this process using randomly shuffled $\beta_p$ rates.

**Estimating the impact of experimental noise.** We generated simulated mRNA and ribosomal footprints abundances for a range of measurement errors. These values, which can be directly inferred from the mRNA-seq and ribosome profiling RPKMs, correspond to the experimental component of the calculations of $\beta_m$ and $\beta_p$.

Each simulated dataset was obtained as follows. Rates $\beta_m$ and $\beta_p$ were first sampled from the dataset produced by [25] for paralog $P_1$ of each of $n$ gene pairs. Then, sets of $\beta_m$ and $\beta_p$ $log_2$-fold changes were sampled randomly from two normal distributions of mean 0 and respective standard deviations $\sigma_{\Delta\beta m}$ and $\sigma_{\Delta\beta p}$. The transcription and translation rates for all paralogs $P_2$ were next computed by applying the selected fold changes to the rates previously sampled for the $P_1$ of each pair. These first steps generate the true $\beta_m$ and $\beta_p$ for the simulated gene pairs, from which noisy experimental measurements were subsequently inferred. From Eq 2, the true $\beta_m$ values can directly be used as (exact) measurements of mRNA abundance $m$, since mRNA decay rate $\alpha_m$ is assumed to be constant. Using Eq 3 and ignoring all constants, equally exact measurements of the abundance of ribosomal footprints $s_i$ for gene $i$ (ribosome profiling RPKMs) can be obtained as: $s_i = \beta_{p,i} m_i$. Following this calculation, experimental variation was added to the $m$ and $s$ measurements for each paralog as Gaussian noise. Relative measurement errors were sampled independently for both properties of each gene from normal distributions with means 0 and respective standard deviations $cv_{\beta m}$ and $cv_{\beta p}$, which are a percentage of the corresponding $m$ or $s$, and added to the exact experimental measurements previously calculated. Apparent $\beta_m$ and $\beta_p$ rates were finally computed from the noisy measurements, and used in the calculation of apparent $log_2$-fold changes, which were themselves compared to the true sampled fold changes.

A slightly modified version of this approach was also used to estimate the combined impact of noise and gene-to-gene variations in mRNA decay. In this case, the initial sampling of $\beta_m$ and $\beta_p$ rates and the corresponding fold changes was accompanied by a draw of experimental mRNA decay rates and $log_2$-fold changes—both taken from one of the datasets previously used [26–29]. Then, when calculating the exact experimental measurements (before the addition of noise), mRNA abundance for gene $i$ was computed as $m_i = \frac{\beta_{m,i}}{\alpha_{m,i}}$. This way, simulated experimental measurements at both levels were impacted by variations in mRNA decay. Once Gaussian noise had been added as previously, apparent $\beta_m$ and $\beta_p$ values were computed under the assumption of invariable mRNA decay rates (using constant $\alpha_m = 5.10 \text{ h}^{-1}$), which were used to obtain the apparent fold changes. These were again compared to the true fold changes initially sampled.

**Assessing the significance of divergence correlations.** We looked at the correlations which could be expected from the fact that mRNA abundance $m$ is used in the calculations of both $\beta_m$ and $\beta_p$. We first considered a scenario in which $m$ and the abundance $s$ of ribosomal footprints are entirely independent. For both variables, pairs of values (for paralogs $P_1$ and $P_2$) were sampled from normal distributions and converted to pseudo rates of transcription and

translation ($\beta_m = m$ and $\beta_p = \frac{s}{m}$), as previously. From these, absolute and signed $log_2$-fold changes were obtained, and then used to compute the two correlations of divergence. A similar approach was used to consider a situation in which $m$ and $s$ are very strongly correlated ($r$ = 0.98), as in the dataset [30] used by [25]. In that case, $m$ and $s$ values for each paralog were sampled simultaneously from a bivariate normal distribution showing such a high correlation. Absolute and signed $log_2$-fold changes were then computed and used to calculate the divergence correlations, as described.

**Measuring the relationship between expression divergence and functional changes.**   To investigate the relationship between the patterns of expression divergence and functional changes within paralog pairs, we introduced a divergence ratio $D$, measuring the bias towards transcriptional changes, and correlated it with proxies of functional divergence. This correlation analysis was also repeated to investigate how the magnitude of protein abundance divergence ($log_2$-fold changes of estimated protein abundance obtained from the $\beta_m$ and $\beta_p$ values; see Eq 5) and functional changes are related.

$$D = log_2 \left( \frac{max\,(\beta_{m,1}, \beta_{m,2}) / min\,(\beta_{m,1}, \beta_{m,2})}{max\,(\beta_{p,1}, \beta_{p,2}) / min\,(\beta_{p,1}, \beta_{p,2})} \right) \tag{4}$$

The matrix of pairwise genetic interaction profile similarities was downloaded (access: 2021-12-17) from TheCellMap.org [73]. Only results from the AllxAll genetic interaction screen were considered. All paralog pairs for which both duplicates were represented in this dataset were kept, leaving 377 gene pairs. When more than one unique mutant had been screened for one gene, the corresponding vectors of pairwise similarity were averaged. For each paralog pair, we kept the mean of the reported similarity (Pearson correlation) between the interaction profiles of genes $P_1$ and $P_2$.

For GO overlap, the GO Slim mappings from the SGD project [74] were downloaded (http://sgd-archive.yeastgenome.org/curation/literature/[2022-01-12]). Annotations from the three ontology levels (Process, Function and Component) were combined and the Jaccard index was computed within each paralog pair as the ratio of the intersection over the union of GO terms.

Amino acid sequences for all *S. cerevisiae* ORFs were downloaded from the SGD project (http://sgd-archive.yeastgenome.org/sequence/S288C_reference/orf_protein/[2022-01-12]). Pairwise global alignments were performed within each of the 409 paralog pairs using BioPython (v 1.80) and the corresponding amino acid identities were computed.

These three proxies of functional divergence were correlated with the divergence ratios $D$ and $log_2$-fold changes of protein abundance described above for each paralog pair using Spearman's $\rho$.

## Minimal model of post-duplication evolution

**Selection on cumulative protein abundance.**   We defined a minimal model of post-duplication evolution based on the idea of quantitative subfunctionalization [23]. Accordingly, selection acts to maintain the cumulative protein abundance of two paralogs near an optimal level.

In accordance with [25], an ancestral gene with a transcription rate $\beta_m$ and a translation rate $\beta_p$ is defined. The resulting steady-state protein abundance is obtained using equation Eq 5 [25], where $\alpha_m$ and $\alpha_p$ are the previously described constants set to median mRNA and

protein decay rates of 5.10 h$^{-1}$ and 1.34 h$^{-1}$ [25].

$$p = \frac{\beta_m \beta_p}{\alpha_m \alpha_p} \tag{5}$$

This gene is also associated with a unique function $W(p)$ of fitness according to protein abundance. This function is assumed to be a parabola of vertex $(p_{opt}, \mu)$, where $p_{opt}$ is a gene-specific protein abundance optimum (in proteins per cell) and $\mu$ is the maximal growth rate of *S. cerevisiae* (0.42 h$^{-1}$ [25]). This function is additionally described by a noise sensitivity parameter $Q$, which measures curvature relative to the value of $p_{opt}$ [25] and is also gene-specific. A higher $Q$ means that fitness decreases more sharply following any relative protein abundance variation away from the optimum (for instance ±5% of $p_{opt}$), and thereby represents a more stringent selection to maintain protein abundance. The three parameters $a$, $b$ and $c$ of the standard form of the parabolic fitness function $W(p)$ are all obtained directly from $p_{opt}$, $\mu$ and $Q$ (see S1 Methods).

Following the duplication of any ancestral gene, two paralogs $P_1$ and $P_2$ are considered. Each inherits the ancestral transcription and translation rates, meaning that $\beta_m = \beta_{m,1} = \beta_{m,2}$ and $\beta_p = \beta_{p,1} = \beta_{p,2}$. The total number of mRNAs synthesized is thus doubled while the translation rate of each transcript remains unchanged, such that protein abundance is also doubled. The function $W(p)$ becomes the function $W(p_1 + p_2)$ of fitness according to cumulative protein abundance for the duplicate pair. Because it seems unrealistic to assume that the new post-duplication protein abundance is perfectly optimal, the optimum of this function is set to $1.87p_{opt}$, such that the gene doubling overshoots the optimal expression level. This value has been selected as it is the smallest multiple which ensures fitness $W(p_1 + p_2) > 0$ immediately after the duplication event, even for the narrowest fitness function (S1 Methods). Apart from the optimum, the other parameters ($Q$ and $\mu$) of the parabolic function do not change following the duplication.

Within this minimal model, mutations affecting the $\beta_m$ and/or $\beta_p$ of a paralog are filtered by natural selection solely according to their effect on fitness $W(p_1 + p_2)$.

**Addition of precision-economy constraints.** To obtain the precision-economy model, we added precision-economy constraints to the minimal model described above. These implement the evolutionary trade-off between the precision and economy of gene expression that was conceptualized by [25]. This involves modifying the fitness calculations to account for stochastic fluctuations of protein abundance (expression noise) and transcription costs.

According to [25], the variance of protein abundance for a singleton gene within a population of isogenic cells can be approximated from the relative contribution of transcription to its expression. This is done according to Eq 6, where $c_{v0}$ is a constant representing the minimal coefficient of variation for protein abundance observed within a clonal population, referred to as noise floor. In *S. cerevisiae*, its value is 0.1 [25].

$$\sigma^2 \approx p^2 \left( \frac{1}{p} + \frac{\alpha_p}{\beta_m} + c_{v0}^2 \right) \tag{6}$$

From this relationship, we have obtained an equation for the variance $\sigma_{tot}^2$ on the cumulative abundance of a protein expressed from two identical paralogous genes, according to the relative expression levels and transcription rates of each of the two gene copies (S1 Methods).

Using this variance $\sigma_{tot}^2$ and the parabolic function $W(p_1 + p_2)$, it is possible to compute fitness while taking into account expression noise. In this case, the fitness $F$ becomes the mean of function $W(p_{tot})$ for a population of cells expressing the paralog pair at expression levels $p_{tot}$

distributed around the mean $P_{tot}$ with variance $\sigma^2_{tot}$:

$$F = E(W(p_{tot})) \tag{7}$$

Because the variance $\sigma^2_{tot}$ of cumulative protein abundance $p_{tot}$ and the function $W$ linking its mean to fitness are known, the populational (mean) fitness $F$ can be computed in an exact manner. The mean of $W(p_{tot})$ can be expressed as equation Eq 8, where $a$, $b$ and $c$ are the parameters of the corresponding parabolic function. Using the definition of variance as the difference between the mean of the squares and the square of the mean, the mean $E(p^2_{tot})$ of the squared cumulative protein abundance can be obtained from variance $\sigma^2_{tot}$ and squared mean cumulative abundance $P^2_{tot}$ (Eq 9). Plugging this expression into Eq 8, The fitness for any combination of expression levels when accounting for expression noise is thus given by Eq 10, where $\sigma^2_{tot}$ depends on the relative contribution of transcription to the expression of the paralog pair (S1 Methods).

$$E(W(p_{tot})) = aE(p^2_{tot}) + bE(p_{tot}) + c \tag{8}$$

$$E(p^2_{tot}) = \sigma^2_{tot} + P^2_{tot} \tag{9}$$

$$E(W(p_{tot})) = a(\sigma^2_{tot} + P^2_{tot}) + bP_{tot} + c \tag{10}$$

For a given protein abundance, the fitness cost of expression varies according to whether translation is done from few or many mRNAs. Following work by [25], the cost of transcription $C$ for two paralogs is calculated from equation Eq 11, where $l_m$ is the length of the pre-mRNA (identical for both copies), in nucleotides, and $c_m$ is the transcription cost per nucleotide, in $nt^{-1}$. In the current model, $l_m$ is considered to be a constant and set to the median yeast pre-mRNA length of 1350 nt [25]. The cost per nucleotide $c_m$ is another constant, which has been estimated as $1.2 \times 10^{-9}\ nt^{-1}$ under the assumption that transcriptional resources are limiting and that any increase in transcription level is done at the expense of other transcripts [25].

$$C = l_m c_m(\beta_{m,1} + \beta_{m,2}) \tag{11}$$

Taking expression noise and transcription costs into account, the population-level fitness $F$ at mean cumulative protein abundance $P_{tot}$ thus becomes the mean of fitness $W$ under protein abundance variance $\sigma^2_{tot}$ minus the penalty $C$ (Eq 12). Within the precision-economy model, mutations are favored or not by selection according to their effect on this fitness $F$.

$$F = E(W(p_{tot})) - C \tag{12}$$

## Simulating the expression divergence of paralogs

We used a sequential fixation approach [44] to simulate the expression divergence of paralogs, thus making the simplifying assumption that mutation rate is low enough for only two alleles (the ancestral state and a mutant) of a given duplicate pair to coexist simultaneously in the population.

**Initialization.** To obtain a set of $n$ paralog pairs, the same number of ancestral singletons are first generated as an array of $n$ rows containing ancestral $\beta_m$ and $\beta_p$ values. Thus, no nucleotide or amino acid sequences are modeled—only their expression phenotypes. Two distinct

groups of ancestral genes are generated depending on whether their evolution is simulated according to the minimal model or to its precision-economy implementation.

Combinations of protein abundance $p_{opt}$—obtained from the reported transcription and translation rates using Eq 5—and $Q$ are first independently sampled $n$ times with replacement from the 4440 individual genes (singletons as well as duplicates) included in the dataset. The full distribution of $Q$ for yeast genes had been inferred beforehand in accordance with [25], using Eq 13. For this sampling, values of $Q$ above the theoretical maximum reported by [25] ($\sim 6.8588 \times 10^{-6}$) are excluded. Combinations of $p_{opt}$ and noise sensitivity resulting in a fitness function curvature below a specified threshold are also filtered out, to avoid cases where a two-fold reduction of cumulative expression immediately after the duplication would be neutral or beneficial (see S1 Methods).

$$Q = c_m \alpha_m l_m \frac{\beta_m}{\beta_p} \tag{13}$$

For the first group of ancestral singletons, used in simulations under the minimal model, transcription and translation rates are set accordingly with Eq 13. It is used to calculate an optimal $\frac{\beta_p}{\beta_m}$ ratio set by precision-economy constraints [25]. The transcription rate $\beta_m$ and translation rate $\beta_p$ which satisfy both this ratio and the optimal expression $p_{opt}$ are then computed. Strictly speaking, an infinity of combinations of $\beta_m$ and $\beta_p$ are optimal for any gene under the minimal model, as long as they result in the specified protein abundance $p_{opt}$. We thus use equation Eq 13 to reproducibly choose one realistic combination for each ancestral singleton. Because the minimal model should not account in any way for the precision-economy constraints on gene expression, $Q$ values are subsequently resampled with replacement from the genomic distribution. As previously, values above the theoretical maximum are excluded and the sampling is repeated if the new $Q$ results in a fitness function curvature below the threshold.

To define the second group of ancestral genes—for simulations under the precision-economy trade-off –, the first combinations of $Q$ and $p_{opt}$ generated are again taken as a starting point. This time, ancestral $\beta_m$ and $\beta_p$ are set to the combination of rates which maximizes fitness $F$ (Eq 12), computed according to gene-specific functions $W(p)$ of fitness according to expression level. This optimization is performed using a differential evolution algorithm—the *differential_evolution* method of the *optimize* suite of the *SciPy* module [75]. Although equation Eq 13 already describes an optimal $\beta_m$–$\beta_p$ pair, the latter may not correspond to the true optimum, as it is restricted to values which result in protein abundance $p_{opt}$. Since the magnitude of stochastic expression fluctuations (noise) scales with protein abundance (Eq 6), the true optimal expression under the precision-economy trade-off may indeed be slightly below the abundance optimum $p_{opt}$.

Once generated, all ancestral genes are duplicated as previously described into two paralogs $P_1$ and $P_2$, both with the ancestral rates $\beta_m$ and $\beta_p$. For each duplicate pair, a new function $W$ ($p_{tot}$) of fitness according to cumulative protein abundance is defined and its optimum is set to the ancestral $p_{opt}$ times $\Delta_{opt} = 1.87$.

**Mutation-selection approach.** The two sets of duplicate pairs thereby generated are then used in a sequential fixation simulation, as previously described. One random mutation is first sampled individually for each of the $n$ paralog pairs from a normal distribution with mean 0 and standard deviation $\sigma_{mut}$. These mutational effects are each assigned randomly to transcription or translation rates according to relative probabilities $P_{\beta m}$ and $P_{\beta p}$, which represent the relative mutational target sizes of the two traits. For instance, $P_{\beta m} = 2$ and $P_{\beta p} = 1$ would be used in a simulation where the mutational target size of transcription is assumed to be twice

that of translation. Once it has been defined as either transcriptional or translational, each mutation is assigned at random to one of paralogs $P_1$ and $P_2$ of the corresponding duplicate pair, which are both equally likely to be mutated. These steps of mutation generation and assignment are done at once for the two sets of simulated gene pairs. As such, the nth paralog couple of both simulations—respectively performed according to the minimal and precision-economy models—receives the exact same series of mutations.

So that their impact on fitness can be assessed, the sampled mutational effects are applied to the transcription or translation rates of the designated gene copies across the minimal model and precision-economy simulations. Epistasis is assumed to be multiplicative, so that every mutational effect is relative to the current trait value as shown in equation Eq 14, where $\delta_m$ and $\delta_p$ respectively represent the transcriptional and translational magnitudes of the mutation. Because a given mutation affects $\beta_m$ or $\beta_p$ but not both simultaneously, at least one of $\delta_m$ and $\delta_p$ is null every time a mutational effect is applied.

$$\beta_{mj} = \beta_{mi} + \delta_m \beta_{mi}$$
$$\beta_{pj} = \beta_{pi} + \delta_p \beta_{pi}$$

(14)

The new transcription and translation rates are used to compute mutant fitness values $F_j$ for each paralog pair of the two simulations, according to the specifications of the minimal and precision-economy models. Thus, while the same mutations are attempted across the two models, they do not necessarily have the same fitness effects in both scenarios.

Mutant fitness $F_j$ is compared to ancestral fitness values $F_i$ computed using the pre-mutation $\beta_m$ and $\beta_p$ of all duplicates and a fixation probability $P_{fix}$ is calculated for each mutation. This is done using a modified Metropolis criterion to accelerate the simulation [34]. Following equation Eq 15, any beneficial or completely neutral mutation is automatically accepted, while deleterious mutations are fixed with a probability which decreases exponentially according to the magnitude of their fitness effect.

$$P_{fix} = \begin{cases} 1 & \text{if } F_j > F_i \\ e^{-2N(log(F_i)-log(F_j))} & \text{if } F_j \leq F_i \end{cases}$$

(15)

Prior to this fixation probability calculation, all fitness values—which are growth rates between 0 and 0.42 $h^{-1}$—are scaled between 0 and 1. The same set of randomly generated floats is used in both simulations to decide whether each mutation is rejected or reaches fixation according to $P_{fix}$. Any mutation resulting in $F_j < 0$ is automatically rejected. In addition, no mutation taking $\beta_m$ or $\beta_p$ to 0 or increasing either above the maximal value reported by [25] is tolerated, irrespective of its fitness effect. Once the fate of each mutation has been established, the new transcription and translation rates of all simulated paralogs are set and this process of mutation-selection is repeated.

Simulations are stopped as soon as they have resulted in a magnitude of protein abundance divergence consistent with what is observed for the extant pairs of yeast paralogs. Following each mutation-selection round, the $log_2$-fold change of protein abundance is calculated for every simulated duplicate pair using Eq 1. Mood's median test is used to compare this distribution to its empirical equivalent for real paralog pairs, computed from estimated protein abundances obtained from reported $\beta_m$ and $\beta_p$ values (Eq 5). Once a p-value > 0.1 is obtained, the simulated protein abundance divergence is considered to have reached a realistic magnitude and the simulation is stopped. This is done separately for the

two models, so that their respective simulations may not be completed in the same number of rounds.

**Implementing asymmetrically distributed mutational effects.** So that mutations increasing or decreasing expression could occur with different frequencies, the mutation-selection framework described above was slightly modified. The normal distribution from which mutational effects are sampled was replaced by a skew normal distribution of mean 0, standard deviation $\sigma_{mut}$ and skewness parameter $\alpha \neq 0$. No changes were made to the steps of mutation assignment and selection.

**Implementing correlated mutational effects.** The sampling of mutational effects was modified to consider a bivariate normal distribution of means 0 and standard deviations $\sigma_{\beta m}$ and $\sigma_{\beta_p}$. Within this new framework, the step of assigning mutations to one level of regulation or the other with relative probabilities $P_{\beta m}$ and $P_{\beta p}$ has to be skipped, because each affects transcription and translation simultaneously. Consequently, mutational target size differences were modeled differently: the magnitudes of transcriptional and translational mutational effects were used instead of their relative probabilities. A larger mutational target size was thus implemented as a higher corresponding standard deviation of effects. For instance, a target size twice larger for transcription than for translation was modeled as $\sigma_{\beta m} = 2\sigma_{\beta p}$.

The precise values of $\sigma_{\beta m}$ and $\sigma_{\beta p}$ were set to ensure a roughly constant expected protein abundance change per mutation across all simulations. To this end, an additional optimization step was added to the initialization of simulations. During this step, a brute-force optimization approach—the *brute* method of the *optimize* suite of the *SciPy* module [75]—is used to find the $\sigma_{\beta m}$ and $\sigma_{\beta p}$ values which verify the desired mutational target size ratio while resulting in the same mean absolute protein abundance change per mutation as a chosen reference $\sigma_{mut}$ in the general framework. From the two resulting standard deviations, the covariance matrix of the bivariate normal distribution is computed, according to a specified correlation coefficient $r_{mut}$ between $\delta_m$ and $\delta_p$.

**Assessment of the simulations.** Once the two simulations are completed, summary statistics are computed for each of them. The resulting distributions of $log_2$-fold changes (Eq 1) in transcription rates, translation rates and protein abundance are compared to their empirical counterparts using the two-sample KS test and Mood's median test. Depending on the simulation runs, this comparison is made with the set of WGD- or SSD-derived true paralog pairs, or both. The two previously described correlations are also computed for the gene pairs obtained from each simulation.

These steps are additionally repeated when considering only gene pairs which would have remained as duplicates even if loss-of-function mutations had been allowed. Summary statistics are thus recalculated for the subset of simulated pairs for which the loss of either paralog would be deleterious (decreases fitness by more than $\frac{1}{N}$).

## Simulation runs

Runs of the simulation script were parallelized on a computing cluster using *GNU parallel* [76]. Except in one instance (Fig 3), all simulations were done in three replicates of 2500 paralog pairs, using the same set of three random seeds throughout. In addition, except when noted otherwise (S13 Fig), the WGD-derived paralogs of *S. cerevisiae* were taken as a reference for the evaluation of the end condition and for the calculation of summary statistics. Most simulations were repeated for the same set of mutational target size ratios ($P_{\beta m}/P_{\beta p} \in \{1/2, 1, 2, 3, 4, 5, 6, 7, 8, 9, 10\}$), unless described differently. The details of each set of simulations are described in the corresponding figure legends and text.

**Identification of the best-fitting standard deviation of mutational effects.** A grid search of $\sigma_{mut}$ and mutational target sizes ratio $P_{\beta m}/P_{\beta p}$ was performed. It was done twice, under the assumption of high ($N = 10^6$) and reduced ($N = 10^5$) selection efficacy. Separately for each value of $N$, the best-fitting $\sigma_{mut}$ was defined as the one which resulted in the smallest observed grand mean KS statistic (mean KS value for $\beta_m$, $\beta_p$ and protein abundance across the three replicate simulations), irrespective of the combination of model (minimal or precision-economy) and $P_{\beta m}/P_{\beta p}$ for which this minimum occurred (see S6 Fig).

**Impact of asymmetrically distributed mutational effects.** Further simulations were done using the modified mutation-selection approach which samples mutations from a skew normal distribution. These simulations, which considered a range of negative values for skewness parameter $\alpha$, were performed at the best-fitting $\sigma_{mut}$ initially identified for WGD-derived paralogs, successively for $N = 10^6$ and for $N = 10^5$.

**Impact of correlations between transcriptional and translational mutations.** The grid search of evolutionary parameters was repeated for the mutation-selection framework adapted to use a bivariate distribution of mutational effects. The range of $\sigma_{mut}$ previously considered were set as reference values, according to which the standard deviations $\sigma_{\beta m}$ and $\sigma_{\beta p}$ were calculated individually for each simulation. This grid search was repeated for the two levels of selection efficacy $N$ (S10 Fig). In each case, the best-fitting reference $\sigma_{mut}$ was identified using the same definition as previously and used in subsequent simulations combining a range of correlation coefficients $r_{mut}$ between the transcriptional and translational effects of mutations (Fig 5, S11 and S12 Figs).

## Supporting information

**S1 Fig. Taking into account variations in mRNA decay when calculating transcription rates does not impact the relationship between transcriptional and translational divergence within yeast paralog pairs.** (A) Correlations between the different sets of transcription rates $\beta_m$ used. The rates originally reported by [25] are identified as "Hausser et al.", while the four others are $\beta_m$ recomputed using the data from [25] and the corresponding set of gene-specific experimental measurements of transcript decay [26–29]. (B) Distributions of relative divergence in transcription and in translation for *S. cerevisiae* paralog pairs by duplication mechanism. Transcription rates have been recalculated using the corresponding set of mRNA decay measurements, while translation rates are the same as in Fig 1. P-values from Mann-Whitney-Wilcoxon two-sided tests are shown. (C) Correlation (Spearman's $\rho$) between the magnitudes of relative divergence in transcription and translation rates across all paralog pairs, using $\beta_m$ rates recalculated when accounting for variations in transcript decay rate. (D) Correlation (Spearman's $\rho$) between the signed relative divergences in transcription and in translation for all gene pairs, when $\beta_m$ is calculated using the corresponding mRNA decay measurements. Each correlation was computed on a duplicated dataset, obtained by calculating the signed $log_2$-fold changes in the two possible orientations for each pair of duplicates. (E) Correlation of the $log_2$-fold changes of protein abundance within paralog pairs estimated from the $\beta_m$ and $\beta_p$ with experimentally measured differences in protein abundance [69, 71, 72]. The estimated protein abundance fold changes were computed using two sets of experimental measurements of protein decay rates [69, 70] and compared to each of four sets of protein abundances. Correlations obtained for the reported $\beta_p$ rates (left) as well as for randomly shuffled $\beta_p$ rates (right) are shown.
(TIF)

**S2 Fig. Experimental noise on mRNA-seq and ribosome profiling measurements is unlikely to explain the predominance of transcriptional changes within yeast paralog**

**pairs.** Transcription rates $\beta_m$ and translation rates $\beta_p$ were computed from simulated noisy measurements for randomized gene pairs. The median ratio of transcriptional over translational changes was calculated across all pairs from the noisy measurements, and compared to the ratio obtained from the underlying true $\beta_m$ and $\beta_p$ (center of the color scale on each heatmap). A ratio of 1 indicates equal magnitudes of transcriptional and translational changes. (A) When the contribution of transcription to expression divergence is equal (middle) or greater (right) than that of translation, noise only decreases the ratio and the relative impact of transcription changes is underestimated. An overestimation of the ratio—and thus of the importance of transcription—can occur when translation divergence dominates (left), but its impact is small. The respective standard deviations of the distributions of $\beta_m$ and $\beta_p$ $log_2$-fold changes ($\sigma_{\Delta\beta m}$ and $\sigma_{\Delta\beta p}$) have been set according to our estimates of relative divergence. When transcription is assumed to dominate (right, as in the dataset from [25]), these standard deviations are $\sim 2.29$, $\sim 1.11$, and vice versa if translational divergence is assumed to be larger (left). Identical contributions of transcription and translation changes (center) are modeled as equal standard deviations resulting in the same total variance. (B) When varying the magnitudes of relative divergence but keeping the transcriptional and translational contributions equal ($\sigma_{\Delta\beta m}$ = $\sigma_{\Delta\beta p}$), noise still mostly results in an underestimation of the contribution of transcription changes. If the the total variance of the $log_2$-fold changes is much larger than empirically (right; $\sigma_{\Delta\beta m}$ = 3), the transcriptional contribution can be overestimated, but the difference is negligible. (C) When translational divergence is assumed to predominate, noise is associated with an overestimation of the contribution of transcription to expression divergence. This overestimation is small—even if the relative magnitude of translation changes is only slightly larger than that of transcriptional divergence (left)—and mostly restricted to situations where noise is larger at the level of mRNA-seq measurements. The summed variance of the $\beta_m$ and $\beta_p$ $log_2$-fold changes is constant across the six heatmaps shown.
(TIF)

**S3 Fig. Experimental noise and mRNA decay variations do not synergize to bias estimates of relative transcriptional and translational divergence within paralog pairs.** Median ratios of $\beta_m/\beta_p$ relative divergence (value of 1 when both are equal) are shown for randomized gene pairs ($n = 10,000$), across a range of noise levels (equal at the transcriptional and translational levels) and experimentally-informed distributions of mRNA decay rate. Transcription and translation $log_2$-fold changes were sampled from distributions with respective standard deviations $\sigma_{\Delta\beta m}$ and $\sigma_{\Delta\beta p}$ for each simulated pair, while $log_2$-fold changes of decay rate were sampled from the corresponding datasets. The "None" column indicates simulated paralog pairs with invariable mRNA decay. The top heatmaps present the apparent ratios, when failing to account for any gene-to-gene variation in transcript decay, while the bottom single-row heatmaps show the true ratios. (A) Using the standard deviations of transcriptional and translational $log_2$-fold changes obtained from the $\beta_m$ and $\beta_p$ rates reported by [25]. The different scenarios of true divergence (e.g. equal magnitudes of transcriptional and translational divergence, middle) correspond to the same standard deviations $\sigma_{\Delta\beta m}$ and $\sigma_{\Delta\beta m}$ as in S2A Fig. (B) Assuming halved variances $\sigma^2_{\Delta\beta m}$ and $\sigma^2_{\Delta\beta p}$ compared to panel A, while still using empirical measurements of mRNA decay.
(TIF)

**S4 Fig. Significant correlations between the transcriptional and translational magnitudes of relative divergence within paralog pairs are expected in the absence of any evolutionary pattern.** Distributions of pseudo $\beta_m$ and $\beta_p$ $log_2$-fold changes have been computed from random variables ($n = 10,000$) mimicking mRNA abundance $m$ and ribosome footprints

abundance $s$. Pseudo $\beta_m$ is set as $m$, while pseudo $\beta_p$ is $\frac{s}{m}$. (A) Expected correlations for the strictly positive (left) and signed (right) $log_2$-fold changes within random pairs when $m$ and $s$ are independent. (B) Correlation between mRNA abundance $m$ and ribosome footprints abundance $s$ in the data from [30] used in the calculation of the reported $\beta_m$ and $\beta_p$ rates for yeast genes [25]. (C) Expected correlations for the strictly positive (left) and signed (right) $log_2$-fold changes when $m$ and $s$ are strongly correlated as seen in B.
(TIF)

**S5 Fig. The predominance of transcriptional divergence is only weakly associated with the level of functional divergence among *S. cerevisiae* duplicated genes.** Correlation of the divergence ratio (left), for which positive values indicate a relatively larger divergence in transcription (Methods), or the global expression $log_2$-fold change (combined effect of transcription and translation; right) with three proxies of functional divergence within paralog pairs, separately for WGD- and SSD-derived ones. Spearman's $\rho$ and p-value were calculated for all duplicate pairs combined. (A) Absolute value of the similarity of genetic interactions profiles within each paralog pair [73]. (B) Overlap (Jaccard index) of GO Slim annotations between paralogs of the same pair. (C) Amino acid identity between the two members of each duplicate pair.
(TIF)

**S6 Fig. Identification of the best-fitting $\sigma_{mut}$ across models and selection efficacies.** Mean Kolmogorov-Smirnov statistics for the comparisons between empirical and simulated relative divergence distributions are shown. Each value is the grand mean for the three parameters ($\beta_m$, $\beta_p$, and $P$) across three replicate simulations of 2500 randomly generated paralog pairs, when compared to yeast WGD-derived paralogs. (A) Minimal model and high selection efficacy. (B) Precision-economy model and high selection efficacy. (C) Minimal model and reduced efficacy of selection. (D) Precision-economy model and reduced selection efficacy. The smallest grand mean KS statistic across A and B is obtained for $\sigma_{mut}$ = 0.025 (minimal model with $P_{\beta m}/P_{\beta p}$ = 3), which is accordingly defined as the most realistic value. For C and D, the minimum value is obtained for $\sigma_{mut}$ = 0.075 (minimal model with $P_{\beta m}/P_{\beta p}$ = 3), which is thus the best-fitting value when a lower selection efficacy is assumed.
(TIF)

**S7 Fig. Replication of relative divergence in transcription, translation and protein abundance across models and levels of selection efficacy for the corresponding best-fitting $\sigma_{mut}$ values.** Both models can generate realistic distributions for all three levels, but the precision-economy model is unable to replicate the observed translational divergence under high selection efficacy. Same simulations as shown in Fig 4, with $\sigma_{mut}$ = 0.025 for $N = 10^6$ and $\sigma_{mut}$ = 0.075 for $N = 10^5$, respectively. (A) Kolmogorov-Smirnov statistics for the three replicate simulations of 2500 randomly generated paralog pairs. (B) Corresponding $log_{10}$-transformed p-values of the Kolmogorov-Smirnov test. The red dashed line indicates the threshold above which differences are not significant. (C) As in A. (D) As in B.
(TIF)

**S8 Fig. Combining a larger mutational target size for translation to the precision-economy trade-off does not produce more realistic translational divergence than the minimal model.** (A) Kolmogorov-Smirnov statistics comparing the fit between empirical (WGD-derived paralogs) and simulated relative divergence distributions for transcription rate ($\beta_m$), translation rate ($\beta_p$) and protein abundance ($P$) for three replicate simulations of 2500 randomly generated paralog pairs when selection efficacy is high ($N = 10^6$) and $\sigma_{mut}$ = 0.025 is postulated. (B) Empirical divergence of WGD-derived yeast paralogs compared with the final

simulated divergence in transcription, translation and protein abundance for simulations under the minimal and precision-economy models when the mutational target size of translation is ten times larger than that of transcription and selection efficacy is high (same simulations as for $P_{\beta m}/P_{\beta p}$ = 1/10 on panel A). (C) Same as A, but for simulations with $N = 10^5$ and $\sigma_{mut}$ = 0.075. (D) Same comparisons as in B, but performed for the simulations with $P_{\beta m}/P_{\beta p}$ = 1/10 shown on panel C. When selection efficacy is high as well as when it is reduced, it is the minimal model which reaches the better fit (lower KS statistic) to the distribution of relative translational divergence.
(TIF)

**S9 Fig. The postulated post-duplication change of optimal cumulative protein abundance has limited influence on the divergence correlations generated from simulations.** Simulated divergence correlations for three replicate simulations of 2500 randomly generated paralog pairs under the minimal and precision-economy models for a sample of mutational target size ratios and a range of post-duplication variation $\Delta_{opt}$ of the cumulative protein abundance optimum, when the efficacy of selection is assumed to be high ($N = 10^6$; $\sigma_{mut}$ = 0.025). A $\Delta_{opt}$ of 2.0 indicates that the doubling of protein abundance resulting from the duplication event is perfectly optimal. All simulations stopped and evaluated according to the empirical expression divergence of yeast WGD-derived duplicate pairs, as in Fig 4. The dashed lines and the shaded areas represent the empirical value of the corresponding correlation and its 95% confidence interval. (A) Correlation between the magnitudes of the $log_2$-fold changes in transcription and translation. (B) Correlation between the signed $log_2$-fold changes in transcription and translation.
(TIF)

**S10 Fig. Identification of the best-fitting standard deviations of mutational effects across models and selection efficacies when bivariate mutational effects are used.** Under this framework, standard deviations $\sigma_{\beta m}$ and $\sigma_{\beta p}$ of transcriptional and translational effects are set by the relative mutational target sizes $P_{\beta m}$ and $P_{\beta p}$, but their precise values are chosen to result in the same mean change of protein abundance per mutation as a reference $\sigma_{mut}$ (shown on the figure) in the univariate implementation (Methods). The mean Kolmogorov-Smirnov statistics for the comparisons between empirical and simulated relative divergence distributions are shown. Each value is the grand mean for the three parameters ($\beta_m$, $\beta_p$, and $P$) across three replicate simulations of 2500 randomly generated paralog pairs, performed and assessed according to the WGD-derived paralogs of yeast. (A) Minimal model and high selection efficacy. (B) Precision-economy model and high selection efficacy. (C) Minimal model and reduced efficacy of selection. (D) Precision-economy model and reduced selection efficacy. The overall lowest grand mean KS statistic across A and B is obtained for a reference $\sigma_{mut}$ of 0.025 (minimal model with $P_{\beta m}/P_{\beta p}$ = 3), indicating that it is the most realistic value. Across C and D, the minimum is obtained for a reference $\sigma_{mut}$ of 0.075 (minimal model with $P_{\beta m}/P_{\beta p}$ = 3), which is thus similarly the best-fitting value when a lower selection efficacy is assumed.
(TIF)

**S11 Fig. The addition of skewness and transcription-translation correlations to the distribution of mutational effects also affects the correlations of divergence generated by the minimal model under conditions of reduced selection efficacy ($N = 10^5$).** (A) Replication of the three distributions of relative divergence (transcription, translation and protein abundance) for a range of mutational effects distribution asymmetry (left) or correlations between the effects of transcriptional and translational mutations (right), as shown by grand mean KS statistics. Asterisks identify instances where all three magnitudes of relative divergence are

realistic for at least one of three replicate simulations ($p > 0.05$, Mood's median test). (B, C) Average final correlation across the three replicate simulations between 1) the magnitudes of transcriptional and translational $log_2$-fold changes (in B) and 2) the signed $log_2$-fold changes in transcription and translation (in C) across the same ranges of mutational effects distribution asymmetry or correlations between the effects of transcriptional and translational mutations. In each case, $\approx$ designates parameter combinations where a correlation coefficient within the 95% confidence interval of the empirical value was obtained for at least one replicate simulation. All corresponding simulations performed as in Fig 5, but for $N = 10^5$ and $\sigma_{mut} = 0.075$. (TIF)

**S12 Fig. The skewness of the mutational effects distribution and the magnitude of the correlation between the transcriptional and translational effects also affect the evolutionary correlations generated by the precision-economy model.** Results are shown for three replicate simulations of 2500 paralog pairs, as in Fig 5 and S11 Fig. (A, B) Average correlation between the magnitudes ($log_2$-fold changes) of relative divergence in transcription and in translation within duplicate pairs, for a range of skewness of mutational effects distribution (left) or of correlations between the $\beta_m$ and $\beta_p$ effects (right). (C, D) Average correlation between the signed magnitudes (signed $log_2$-fold changes) of transcriptional and translational relative divergences along the same ranges of skewness (left) or transcription-translation mutational correlations (right). Across all heatmaps, the symbol $\approx$ identifies parameter combinations where a correlation coefficient within the 95% confidence interval of the empirical value was obtained for at least one replicate simulation. (TIF)

**S13 Fig. The minimal and precision-economy models offer some replication of the divergence patterns of SSD-derived yeast paralogs, but with a lower overall agreement.** The grand means of Kolmogorov-Smirnov statistics (across $\beta_m$, $\beta_p$ and $P$) for three replicate simulations of 2500 randomly generated paralog pairs are shown. Testing of the end conditions of the simulations and comparisons of the resulting divergence patterns were both done according to the SSD-derived duplicate pairs of *S. cerevisiae*. (A) Minimal model and high selection efficacy. (B) Precision-economy model and high selection efficacy. (C) Minimal model and reduced efficacy of selection. (D) Precision-economy model and reduced selection efficacy. (TIF)

**S14 Fig. Simulations under both models generate realistic distributions of genes in the expression space.** Comparisons of the empirical distribution of yeast genes in transcription ($\beta_m$) and translation ($\beta_p$) rates [25] with the distributions of simulated paralogs obtained from the combination of three replicate experiments of *in silico* evolution with 2500 randomly generated duplicate pairs (selected simulations from Fig 4). The simulations resulting in the best-fitting patterns of expression divergence (as previously assessed in S6 Fig) are shown in each case. The dashed line represents the estimated maximal translation rate [25], while the diagonal is the boundary of the depleted region defined by [25]—below which only 1% of all yeast genes are found. (A) Comparisons with simulations made under the minimal model, under assumptions of high (left; $\sigma_{mut} = 0.025$ and $P_{\beta m}/P_{\beta p} = 3$) and reduced selection efficacy (right; $\sigma_{mut} = 0.075$ and $P_{\beta m}/P_{\beta p} = 4$). (B) Comparisons with simulations made under the precision-economy model for both levels of selection efficacy (left: $\sigma_{mut} = 0.025$ and $P_{\beta m}/P_{\beta p} = 1$; right: $\sigma_{mut} = 0.075$ and $P_{\beta m}/P_{\beta p} = 1/2$). (TIF)

**S15 Fig. Qualitatively similar results are obtained when only the duplicates which are the most likely to be retained for an extensive period of time are considered.** All gene loss

events which would have been tolerated by selection at the end of each simulation were performed prior to the calculation of the summary statistics shown. Results for three replicate simulations of 2500 paralog pairs, stopped and evaluated according to yeast WGD-derived duplicates, are shown as previously. (A) Grand means of the Kolmogorov-Smirnov statistics for $\beta_m$, $\beta_p$, and $P$ across ranges of standard deviations $\sigma_{mut}$ and mutational target size ratios, as in S6 Fig). (B) Mean KS statistics and divergence correlations across the range of mutational target size ratios for three replicate simulations performed at the best-fitting standard deviations of mutational effects ($\sigma_{mut} = 0.01$ for $N = 10^6$, and $\sigma_{mut} = 0.05$ for $N = 10^5$), as in Fig 4B–4D. Across the latter panel, patterns very similar to those observed in the corresponding figure of the main text are obtained, but the fit to the empirical distributions is slightly reduced (increased mean KS statistics).
(TIF)

**S1 File. Animation of the divergence of transcription and translation under the minimal model of post-duplication evolution.** Mutations cause small changes in the transcription and translation rates of paralogs $P_1$ and $P_2$. Selection to maintain cumulative expression however maintains their total protein abundance at an approximately constant level.
(GIF)

**S1 Methods. Extended methods.**
(PDF)

## Acknowledgments

We thank all Landry lab members, and especially Angel Cisneros, Philippe Després and Johan Hallin, for insightful journal club discussions which led to this project.

## Author Contributions

**Conceptualization:** Simon Aubé, Christian R. Landry.

**Formal analysis:** Simon Aubé, Lou Nielly-Thibault.

**Funding acquisition:** Christian R. Landry.

**Investigation:** Simon Aubé.

**Methodology:** Simon Aubé, Lou Nielly-Thibault.

**Resources:** Christian R. Landry.

**Supervision:** Christian R. Landry.

**Visualization:** Simon Aubé.

**Writing – original draft:** Simon Aubé, Lou Nielly-Thibault, Christian R. Landry.

**Writing – review & editing:** Simon Aubé, Lou Nielly-Thibault, Christian R. Landry.

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
