## [Decision Letter · Decision Letter 0]

20 Mar 2023

Dear Dr Aubé,

Thank you very much for submitting your Research Article entitled 'Evolutionary trade-off and mutational bias could favor transcriptional over translational divergence within paralog pairs' to PLOS Genetics.

The manuscript was fully evaluated at the editorial level and by independent peer reviewers. The reviewers appreciated the attention to an important topic but identified some concerns that we ask you address in a revised manuscript.

We therefore ask you to modify the manuscript according to the review recommendations. Your revisions should address the specific points made by each reviewer.

Yours sincerely,

Justin C. Fay

Academic Editor

PLOS Genetics

Bret Payseur

Section Editor

PLOS Genetics

The manuscript is to be accepted, but the reviewers did bring up two suggested changes which you may consider. All three of the reviewers also brought up the notion that the manuscript would be easier to read and digest if some of the methodological details were moved out of the results. On the one hand I think it's hard to read papers if one has to continually reference the supplement, but on the other hand two of the reviewers felt the modelling details made it hard to digest. I believe this decision on style is up to you (authors), but it is worth considering some amendment given this is a consensus view.

Reviewer's Responses to Questions

**Comments to the Authors:**

Reviewer #1: The authors have addressed all the points that I had raised in my previous review in a satisfactory manner. Especially, their response to my comments about the noise of measurements for transcription vs translation fully addresses my initial concerns.

While this might not be my place, I still would like to point out something from the (long!) response to reviewer #2. In their first point, reviewer #2 mention the possibility that a majority of synonymous mutations might impact gene expression levels ("Indeed, Shen et al., (2022) showed synonymous mutations are often non-neutral. This creates additional complications as mutations to more common codons may increase expression of the protein while reducing the apparent translation rate from Ribo-seq studies"). There is a great deal of skepticism in the field about these (surprising, to say the least) results and it has been suggested that this observation was an artefact (see: https://www.biorxiv.org/content/10.1101/2022.07.14.500130v1.abstract)

I also agree with reviewer #2 that some sections of the paper (especially the modeling part) are sometime difficult to follow, but I don't really have any solution to offer. It seems to me that the complexity is just inherent to the question being studied and the approach (complex simulations).

I do note that the authors had to re-run most of their simulations and had to change a few parameters (and correct a few mistakes, notably using log base 10 instead of natural log). While some results are slightly affected, the fact that the main result remains unchanged makes me quite confident that the conclusions reported in this paper are robust.

Reviewer #2: The resubmitted version of Aubé et al’s manuscript is somewhat clearer and easier to follow than the original submission, though I still find some parts of it rather tough going. I can see that the authors made some effort to make the manuscript more accessible to generalist readers, but perhaps they could go further still.

It is a solid and careful piece of work that develops new methods for analysing how transcription and translation can diverge as duplicated genes evolve. It is perhaps unfortunate that both of the models the authors investigated turned out to be plausible, which makes the “take-home” message of the manuscript quite complicated. We don’t often see the word “could” in a manuscript title, and this is an indication of the careful nature of the study.

My only comment is about the cartoons in Figure 2A (left, top). For clarity, shouldn’t the Y-axes of the 2 small graphs be labeled “Precision” and “Economy” rather than “Fitness”? (Fitness is not mentioned in the text on lines 179-192 or the legend to Fig 2A).

Reviewer #3: The revised manuscript is much improved over the original submission. I appreciate the authors extensive work in testing alternative complications to their modeling approach. I feel the content is a good addition to the literature on the topic. My one remaining concern is that the manuscript is very challenging to read, and should be made more concise. I think the main problem is that there is an extraordinary amount of detail in the main text regarding the various parameters used for modeling, rather than focusing on the results of the modeling. In this regard, the manuscript would be easier to read if the authors could please move modeling methods-related material either to the methods section or to a supplementary methods section.

Minor comment - I'm not sure how PLoS journals handle images made on Biorender, and if you would be better off acknowleding that in the acknowledgement section than in figure legends.

**Have all data underlying the figures and results presented in the manuscript been provided?**

Reviewer #1: Yes

Reviewer #2: Yes

Reviewer #3: Yes

PLOS authors have the option to publish the peer review history of their article (what does this mean?). If published, this will include your full peer review and any attached files.

Reviewer #1: **Yes: **Jean-Francois (Jeff) Gout

Reviewer #2: No

Reviewer #3: No

---

## [Editor Report · Decision Letter 1]

21 Apr 2023

Dear Dr Aubé,

We are pleased to inform you that your manuscript entitled "Evolutionary trade-off and mutational bias could favor transcriptional over translational divergence within paralog pairs" has been editorially accepted for publication in PLOS Genetics. Congratulations!

Yours sincerely,

Justin C. Fay

Academic Editor

PLOS Genetics

Bret Payseur

Section Editor

PLOS Genetics

Comments from the reviewers (if applicable):

**Data Deposition**

http://datadryad.org/submit?journalID=pgenetics&manu=PGENETICS-D-23-00087R1

**Press Queries**

---

## [Editor Report · Acceptance letter]

23 May 2023

PGENETICS-D-23-00087R1 

Evolutionary trade-off and mutational bias could favor transcriptional over translational divergence within paralog pairs 

Dear Dr Aubé, 

We are pleased to inform you that your manuscript entitled "Evolutionary trade-off and mutational bias could favor transcriptional over translational divergence within paralog pairs" has been formally accepted for publication in PLOS Genetics! Your manuscript is now with our production department and you will be notified of the publication date in due course.

With kind regards,

Anita Estes

PLOS Genetics

On behalf of:
